

# Patterns and drivers of organic matter decomposition in peatland open-water pools

Julien Arsenault[1,2], Julie Talbot[1,2], Tim R. Moore[3], Klaus-Holger Knorr[4], Henning Teickner[4], Jean-François Lapierre[2,5]

[1]Département de Géographie, Université de Montréal, Montréal, H2V 0B3, Canada
[2]Groupe de Recherche Interuniversitaire en Limnologie (GRIL), Montréal, H2V 0B3, Canada
[3]Department of Geography, McGill University, Montréal, H3A 0B9, Canada
[4]Ecohydrology and Biogeochemistry Group, Institute of Landscape Ecology, University of Münster, Münster, 48149, Germany
[5]Département des Sciences Biologiques, Université de Montréal, Montréal, H2V 0B3, Canada

*Correspondence to*: Julien Arsenault (julien.arsenault.1@umontreal.ca)

**Abstract.** Peatlands pools are unvegetated, inundated depressions that cover up to 30% of the surface of many temperate and boreal peatlands and that are net carbon (C) sources within C-accumulating ecosystems. The emission of carbon dioxide ($CO_2$) and methane ($CH_4$) from peatland pools comes from the degradation of organic matter (OM) that comprise the surrounding matrix. It is, however, not clear how decomposition rates in pools, which define their function and distinguish them from other aquatic ecosystems, vary spatially and what mechanisms drive these variations. We quantified rates of OM decomposition from fresh litter at different depths in six pools of distinct morphological characteristics in a temperate ombrotrophic peatland using litterbags of *Sphagnum capillifolium* and *Typha latifolia* over a 27-month period and measured potential $CO_2$ and $CH_4$ production of pool sediments in laboratory incubations. Rates of decomposition were faster for *T. latifolia* than *S. capillifolium* and, overall, faster at the pool surface and decreased with increasing depth. Pool sediment chemistry was variable among pools and drove the production of $CH_4$ and $CO_2$ from sediments, with decreasing $CO_2$ production with increasing OM humification and decreasing $CH_4$ production with increasing nitrogen-to-phosphorus ratio. Both $CH_4$ and $CO_2$ production from pool sediments were higher in the 1 m deep pools, but similar in the shallow < 1 m and the > 1.5 m deep pools. Our results show that OM decomposition in peatland pools is highly variable and related to OM chemistry, but decomposition depends primarily on the environmental conditions in which it occurs, with differences in both fresh litter and pool sediment decomposability as a function of decreasing $O_2$ concentrations, light, and temperature with increasing depth in the pools.

## 1 Introduction

Peatlands are among the largest carbon (C) reservoirs on the planet, storing twice the C found in forest soils (Loisel et al., 2021; Pan et al., 2011) but covering only 3% of the Earth's surface (Xu et al., 2018). The positive C balance of peatlands is the result of slower organic matter (OM) decomposition rates compared to net primary production in response to anoxic conditions in the cool and water-saturated soils, to functionally limited decomposer communities, and to plant species that are



slow to decompose (Rydin and Jeglum, 2013). Open-water pools that develop in many temperate and boreal peatlands are small (often < 2000 $m^2$ and < 2 m deep; Arsenault et al., 2022) and generally have a negative C balance because of distinct environmental conditions (e.g. higher oxygen – $O_2$ – availability and warmer temperature) compared to the surrounding soils (Foster and Glaser, 1986) that lead to faster OM decomposition than production (Pelletier et al., 2015). Consequently, peatlands

that have pool covers greater than 37% may be net sources of C to the atmosphere (Pelletier et al., 2014).

Within a single peatland, pool morphology (i.e. depth, area, shape) and OM quality may be heterogeneous (Arsenault et al., 2019), which influences the biogeochemical patterns of pools and the magnitude of the processes that control them. For example, shallow pools (< 1 m) have higher concentrations in dissolved organic C (DOC), total nitrogen (TN), $CO_2$ and $CH_4$, and lower pH than deep pools  (> 1.5 m) (Arsenault et al., 2018). Moreover, in a spatiotemporal study of peatland pool

biogeochemistry, Hassan et al. (2023) found that dissolved OM (DOM) concentrations and composition drove intra- and inter-regional patterns in pool carbon dioxide ($CO_2$) and methane ($CH_4$) dynamics. Assuming most of the DOM processed in temperate and boreal peatland pools originates from the surrounding peat and vegetation (Prijac et al., 2022), and with regards to the role pool morphology exerts on biogeochemical patterns and processes, OM degradation rates may vary spatially in response to different environmental conditions. It is, however, unclear how environmental heterogeneity affects decomposition

processes in peatland pools and, more broadly, how it affects the overall capacity of peatlands to sequester C.

Spatial variations in decomposition and C emission rates from pools may emerge as a function of pool depth and hydrological connectivity with the surrounding peat soil. Pools are more oxygenated, warmer and light penetrates deeper in their profile than in the surrounding peat, but $O_2$ concentrations, temperature and light intensity reaching the sediments decrease with increasing pool depth (Arsenault et al., 2018). Degree of anoxia, temperature and luminosity are major drivers of OM decay

in aquatic ecosystems as they control microbial activity of production and decomposition (Pace and Prairie, 2005). Then, OM decomposition rates may be higher in shallow than in deeper pools because of increased microbial activity at the water-peat interface.

Decomposition rates in aquatic and terrestrial ecosystems are also related to the chemical and physical composition of the substrate. For example, in northern peatlands, decay rates ($k$) of litter ranged, over up to six years, from 0.02 $yr^{-1}$ for some

*Sphagnum* species to 0.25 $yr^{-1}$ for vascular plants and > 0.9 $yr^{-1}$ in soft-tissued plants such as *Maianthemum* (Moore et al., 2007). Spatial variations in peat-forming vegetation composition affect the chemistry of the peat and potentially that of the litter that falls in the pools (e.g. Kaštovská et al., 2018). Peat chemistry also varies in depth, with deeper layers of peatland soils being generally depleted in phosphorus (P) and enriched in nitrogen (N) and aromatic organic C molecules that are more recalcitrant to decomposition compared to the surface layers (Cocozza et al., 2003; Wang et al., 2015). It is thus possible that

OM decomposition rates in pools also vary within a peatland because of vegetation heterogeneity and changing substrate quality with increasing pool depth. However, to our knowledge, the relationships between the physical conditions of temperate





peatland pools and the chemical properties of decaying OM, and the control they exert on decomposition rates in peatland pools, have never been studied.

The litterbag technique is a commonly used method to measure monthly to multi-year decomposition rates in aquatic (Tonin
et al., 2018) and terrestrial (Karberg et al., 2008) ecosystems. In peatlands, litterbags have been used to compare decomposition rates of peat-forming plants in different environmental conditions, ranging from anoxic peat layers to oxygenated beaver pond water (e.g. Moore et al., 2007). While litterbags provide insights on environmental controls of OM decomposition, they can hardly account for variations in material molecular composition such as observed along peat profiles as both peat quality and environmental conditions change with depth. Laboratory incubations of peat and pool sediments, where decomposition of
different materials can be measured under controlled conditions (Schädel et al., 2020), can provide this extra information on the degradability of OM of different molecular composition. Hence, a combination of *in situ* and laboratory experiment approaches are needed to isolate the mechanisms controlling OM decomposition in peatland pools.

Pools are often overlooked in peatland studies (Loisel et al., 2017), hence little focus has been put on the mechanisms controlling the rates at which decomposition processes occur in such features. Hence, the relationships between the structure
of peatland pools and OM processing and, more generally, the role pools play in the C budget of a peatland, remain unclear. In this context, this study aimed at providing a mechanistic understanding of OM decomposition in peatland pools. Specifically, the objectives were to i) quantify rates of similar, fresh OM litter decomposition in pools of different morphological and chemical properties; ii) measure the degradability of individual pool sediments under controlled conditions; and iii) assess the role OM chemistry plays on decomposition rates for both litter and sediments. Using the litterbag technique for a 27 month *in*
*situ* experiment and laboratory incubations of pool sediments, we measured rates of decomposition of OM in and from peatland pools (Figure 1) and characterized spatiotemporal changes in OM composition.

## 2 Methodology

### 2.1 Site description

*In situ* experiments and organic matter (OM) samples for degradation experiments were collected in Grande plée Bleue (GPB),
a 15 km$^2$ mostly undisturbed ombrotrophic peatland located 12 km southeast of Québec City, Canada (46°47'N, 71°03W, altitude ~88 m). Its climate is cool continental, with a mean annual air temperature of 4.8 °C and a mean annual precipitation of 1066 mm (206 mm of which fell as snow) for the 1991-2020 interval at the nearest meteorological station (4.2 km from the study site) (Canadian Climate Data - Saint-Michel Station, 2022). The peatland was initiated over marine sediments of the Goldthwait Sea 9500 years ago and expanded laterally through the paludification of surrounding forests (Lavoie et al., 2012).
Ombrotrophic conditions prevailed for the last 8300 years, and peat depth reaches 5 m in the center of the peatland (Lavoie et al., 2012). Vegetation composition varies in the peatland, with some areas dominated by *Sphagnum* mosses, small ericaceous shrubs (e.g. *Kalmia angustifolia, Chamaedaphne calyculata*) or graminoid (e.g. *Carex* spp., *Eriophorum virginicum* and





*Rhyncospora* spp.), some by continuous shrub covers (e.g. *K. angustifolia, Rhododendron* spp. and *Gaylussacia baccata*) underlain by *Sphagnum* mosses, and others by dense coniferous trees (e.g. *Picea mariana* and *Larix laricina*).

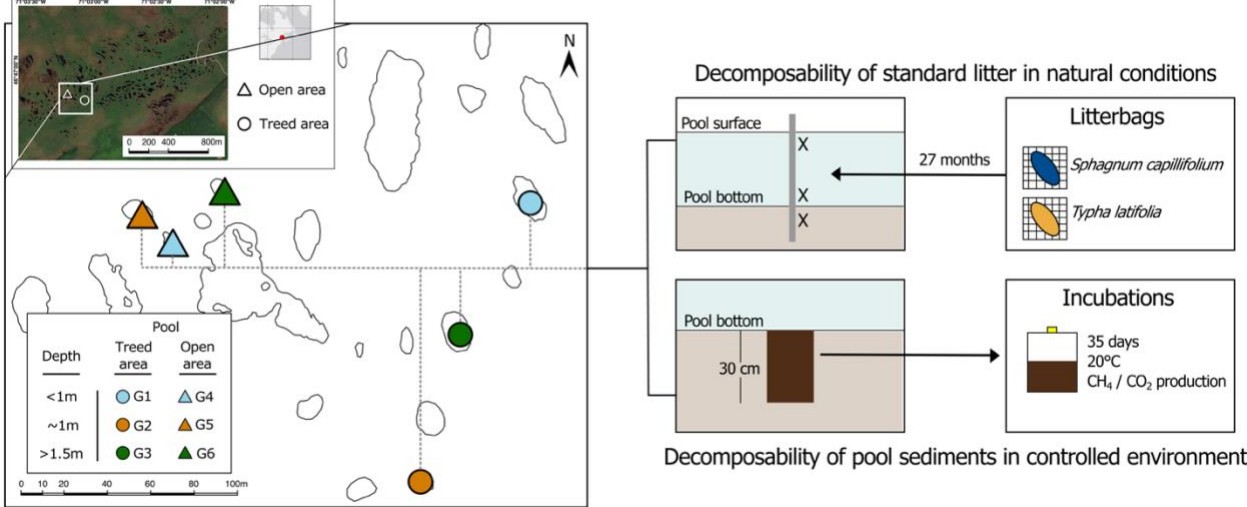


Figure 1. Conceptual framework of the experiment. Six pools of different depth categories were selected in two areas of the peatland characterized by distinct vegetation assemblages. We installed litterbags of *Sphagnum capillifolium* and *Typha latifolia* over a period of 27 months at the surface, near the bottom and in the sediments of each pool to measure the decomposability of standard fresh litter under natural conditions. We conducted incubations on sediment samples collected at the center of each pool to measure the decomposability of pool bottom material under controlled settings and assessed pool sediment chemistry and changes in litter chemical composition after 27 months. The satellite photo of the site is from DigitalGlobe WorldView-2 imagery.

The southern and central parts of the peatland are characterized by the presence of > 600 pools of different shape, area (10-2350 $m^2$, mean 381 $m^2$; n = 152) and depth (15-219 cm, mean 93 cm; n = 158). For this study, we selected six pools: three pools located in an open area with vegetation mainly composed of extensive *Sphagnum* moss mats and small, sparse *C. calyculata* and *K. angustifolia* shrubs, and three pools in a treed area with *P. mariana* (height > 2 m) dominating the canopy (Figure 1). In each area, the three pools that we selected were of different depths (shallow, < 1 m; average, ~ 1 m; deep, > 1.5 m). In each pool, we installed litterbags, took samples of the sediment and underlying peat, and collected water and gas samples.



## 2.2 Litterbags

To assess the effect of pool chemical and physical characteristics on decomposition rates, we installed litterbags of *Typha latifolia* leaf and stem tissues (collected at Mer Bleue peatland, Ontario, and cut in 10 cm lengths) and *Sphagnum capillifolium* (capitulum and first 5 cm of the stem, harvested *in situ*) in each of the 6 pools and in the surrounding peat. Upon collection,
litter was oven-dried (40 °C). Litterbags were made of fibre glass mosquito netting with a 1 x 1 mm mesh size. Bags were numbered, weighed, filled with approximately 2 g of dry material, then weighed again to obtain the initial mass of litter. Material was then rewetted using deionized water prior to field installation.

In June 2019, we installed triplicates of both types of litter at three depths (at the surface, 15 cm above the pool bottom, and 15 cm below the sediment-water interface) in each pool. Surface litterbags were attached to a buoy. To reach deep locations,
bags were fixed on wood stakes that were inserted in the pool sediments. In the two areas, we also installed triplicates of litterbags of the two litter types in the unsaturated surface layer of the peat surrounding the pools (5 cm under the peatland surface) to compare decomposition rates between the acrotelm and the pools.

Litterbags were retrieved five times over the course of three growing seasons. A first set of bags was retrieved from each pool and from the peat in the two zones at the end of the 2019 growing season (4 months after installation). The other bags were
retrieved in the spring and fall 2020 (respectively 11 and 16 months after installation) and again in the spring (23 months) and fall 2021 (27 months). Upon collection, litterbags were stored at -20°C until analysis. After thawing at 4°C, bags were carefully washed using deionized water and tweezers to remove any alien material. Remaining litter was oven-dried (40°C) for 72 h, then weighed, and percent of original mass was calculated. We then milled the litter and analyzed for molecular composition of C compounds to assess changes between the original material and material that has been exposed to environmental
conditions for 27 months in the pools.

## 2.3 Incubations

To test for differences in peat composition and degradability among pools, we collected samples of bottom material at the center of the six studied pools. Using a Kajak-Brinkhurst corer, in spring 2021, we sampled the first 30 cm of pool sediments to reach both limnic sediments and the underlying peat. We transferred each core in clean low-density polyethylene bags and
homogenized the material. Back in the lab, we drained the samples of excess water using a cheese cloth (24 h at 4°C). We then brought the samples to room temperature for another 24 h before we started the incubations.

From each pool sediment sample, we took five subsamples for incubations. At the initial time, we half-filled 250 mL glass jars with the drained, homogenized pool bottom subsample, added 30 mL of deionized water to measure DOC production after incubations and sealed the jars with lids equipped with stopcock valves attached to rubber tubes for headspace sampling. Jars
were kept at 20°C with no direct sunlight for the entire incubation time. For each set of pool subsample, we also added a sealed, empty jar to measure ambient air gas concentrations.



Five-milliliter gas samples were taken from the headspace of each jar using a syringe at initial time (T = 0 day) and after T = 0.5, 1, 2, 3, 5, 7, 11, 14, 21, 29 and 35 days. Gas sampled in the headspace was replaced with ambient air to add $O_2$ in the jars and mimic natural $O_2$ production at the water-sediment interface in the pools (Arsenault et al. 2018). Gas samples were rapidly

analyzed for $CH_4$ and $CO_2$ concentrations on a Shimadzu GC-14 gas chromatograph, equipped with a flame ionization detector for $CH_4$, and a methanizer with a 0.7 m column of Ni-reduced catalyst for $CO_2$. Concentrations were corrected to account for the addition of ambient air at every sampling time.

At the end of the incubations, material from each jar was drained to collect water, was oven-dried for 5 days at 40°C and weighed. We also dried the remaining original material at the same time. Because we were unable to retrieve more than 5 mL

of water from any jar, we bulked the incubation water samples per pool and analyzed for DOC and TDN concentrations on a Shimadzu TOC-V total organic carbon analyzer after acidification with HCl (ASTM methods D7573-09 and D8083-16). All dried material was milled and analyzed for molecular composition to characterize differences in sediment material between pools.

We calculated $CH_4$ and $CO_2$ production rates in each jar from between each sampling as the linear change in gas concentration

in the headspace over time, allowing for changes from replacing headspace with ambient air (Kim et al., 2015). We measured ambient $CO_2$ and $CH_4$ concentrations and estimated the post-sampling concentration as the new concentration in the jars until the next sampling by taking into account the addition of 5 mL of ambient air in each jar. Production rates from the five replicates were averaged and reported relative to the dry mass of C. We estimated the 35-days total production of $CH_4$ and $CO_2$ by summing the rates of production in a sampling interval multiplied by the number of days between samplings. Rates

were treated as production when > 0 and as consumption when < 0. Rates were expressed in µg $CO_2$ g $C^{-1}$ day$^{-1}$ and in µg $CH_4$ g $C^{-1}$ day$^{-1}$.

## 2.4 Litter and peat chemistry

To account for temporal changes in fresh litter chemistry and spatial variations in pool sediment chemistry, we analyzed final and initial litter and sediment elemental contents (C, N, P, K, Ca, Na, K, Mg), isotope signatures $\delta^{13}C$ (‰ V-PDB), $\delta^{15}N$ (‰

AIR nitrogen), and absorbance of mid infrared radiation in the range 650 to 4000 cm$^{-1}$ providing information on the relative abundances of molecular structures.

Concentrations and isotopic composition of C and N were measured with an elemental analyzer (EA 3000, Eurovector, Pavia, Italy) coupled to an isotope ratio mass spectrometer (IRMS; Nu Horizon, NU Instruments, Wrexham, UK), using certified reference materials and working standards. All other elemental contents (P, K, Ca, Na, K, Mg) were measured on a wavelength

dispersive X-ray fluorescence spectroscope calibrated with certified and own reference standards (WD-XRF; ZSX Primus II, Rigaku, Tokyo, Japan). To this end, 500 mg of ground, powdered sample were pressed to a pellet with 13 mm diameter (without pelleting aids) at a load of approximately 7t.



Mid infrared spectra (MIRS) were measured on 13 mm diameter pellets pressed from a mixture of 2 mg of milled sample material and 200 mg of KBr (FTIR grade, Sigma Aldrich) on a transmission Fourier transformed infrared spectrometer (Cary
660 FTIR spectrometer, Agilent) in the range 600 to 4000 cm$^{-1}$ at a resolution of 2 cm$^{-1}$ and by averaging 32 scans per spectra. Prior to each measurement, the optical path was purged for 180 s with synthetic dry air. Spectra were background corrected by subtracting a KBr background and converted to absorbance.

In a next step, the spectra were exported to R version 4.2.0 (R Core Team, 2022) and the following preprocessing steps were applied using the 'ir' package version 0.3.0 (Teickner, 2023): (1) Baseline correction using a convex hull-based algorithm in
the 'hyperSpec' package version 0.100.0 (Beleites and Sergo, 2021), (2) clipping of the first and last 5 cm$^{-1}$ and a second baseline correction step to remove artifacts from the first baseline correction step, (3) normalization (so that all intensity values of a spectrum sum to one). These preprocessing steps are commonly applied heuristic procedures to reduce effects such as internal reflectance not directly related to the relative abundance of molecular structures in the sample.

The preprocessed MIRS were analyzed with the 'irpeat' package version 0.2.0 (Teickner and Hodgkins, 2022) by computing
the four humification indices defined in Broder et al. (2012); ratios of the intensities at 1420 (OH and CO of phenols, CH$_2$ and CH$_3$ groups, C–O–H in-plane bending), 1510 (C=C stretching of aromatics, N–H bending and C–N stretching in proteins), 1630 (NH$_2$ bending and C=O stretching in proteins, aromatic C=C and COO$^-$ stretching), and 1720 cm$^{-1}$ (C=O stretching in carbonyls, carboxyls, esters) to the intensity at 1090 cm$^{-1}$ (attributed mainly to C–O stretching of cellulose in absence of silicates) (Broder et al., 2012; Parikh et al., 2014; Stuart, 2004).

**2.5 Pool water chemistry**

At each litterbag retrieval, we collected water samples from each pool. Samples were taken 20 cm below the surface at the center of the pools and analyzed for dissolved organic carbon (DOC), total nitrogen and phosphorus (TN and TP), nitrate (NO$_3$), ammonium (NH$_4$), phosphate (PO$_4$), pH, and absorbance (A$_{254}$) to calculate Specific UV absorbance at 254 nm (SUVA$_{254}$) and estimate DOC aromaticity (Weishaar et al., 2003). During summer 2021, we measured methane (CH$_4$) and
carbon dioxide (CO$_2$) fluxes in each of the six pools using a Li-Cor 8200-01S Smart Chamber coupled to a Li-Cor LI-7810 analyzer. At the same time, we measured pool dissolved CH$_4$, CO$_2$ and N$_2$O concentrations using the headspace technique. Dissolved gas samples were analyzed in the lab on a Shimadzu GC-14 gas chromatograph and concentrations in the pools were obtained using Henry's law for the headspace concentrations. Barometric pressure, wind speed and air temperature were measured *in situ* using a Kestrel 4000 weather meter. Pool water temperature was measured with a DO200A YSI probe.
Laboratory analyses were performed as described in Arsenault et al. (2018).

**2.6 Statistical analyses**

We first determined the differences and similarities in pool morphometry and water chemistry among the six studied pools using Kruskal-Wallis rank and Dunn post-hoc tests for non-normally distributed variables. We measured pool depth and water





chemistry at all retrievals, and we used the average to compare the pools. We did not compare pools with regard for variables
for which we only had one sampling time (e.g. area, dissolved $CH_4$, $CO_2$ and $N_2O$ concentrations, and $CH_4$ and $CO_2$ fluxes).

To determine litterbag decomposition over time, we calculated decay rates ($k$ value) by computing a single-exponential linear regression model ($ln[y] = a - kt$) based on the average percentage of mass remaining ($y$) of triplicates of litter at each retrieving time, with $a$ = intercept, $k$ = decay constant, and $t$ = time in years (Trofymow et al., 2002). After testing for homogeneity of variances, we used two-way ANOVAs to compare $k$ among pool, depth and litter types, and account for the possible
interacting effects of pool characteristics (i.e. morphology and chemistry) and depth of incubation on decomposition. To determine the effect of water chemistry on decomposition rates, we used generalized linear models (GLMs) with gamma distribution and log-link function of the percentage of mass remaining in the bags at the surface and at the bottom of the pools at each retrieving time with water chemistry measured at the same time. We also used GLMs (gamma distribution, log-link function) and Spearman's rank correlations for non-normally distributed variables to quantify the relationship of
$k$ and depth of incubation and to assess the influence of $k$ on final litter material chemistry.

We used Kruskal-Wallis rank and Dunn post-hoc tests for non-normally distributed variables to compare differences in $CO_2$ and $CH_4$ production and sediment chemical properties among pools and between the treed and open areas of the peatland. We also used Spearman's rank correlations and a principal component analysis (PCA) to explore the relationships of $CO_2$ and $CH_4$ total production and sediment chemistry.

All statistical analyses were performed on R, version 4.1.2 (R Core Team, 2022).

## 3 Results

### 3.1 Decomposability of standard litter in natural conditions

#### 3.1.1 Effects of litterbag position on decomposition rates

After 27 months in the pools, *T. latifolia* was more degradable than *S. capillifolium*, and decomposition rates were faster at the
surface of the pools than at depth (Table 1). In particular, decomposition rate constants derived from single-exponential linear regression models for *T. latifolia* ranged from 0.10 to 0.19 $yr^{-1}$ at the surface of the pools and from 0.03 to 0.08 $yr^{-1}$ at depth (Table 1). For *S. capillifolium*, decomposition rate constants ranged from 0.05 to 0.11 $yr^{-1}$ at the surface of the pools and from 0.04 to 0.09 $yr^{-1}$ at depth (Table 1). In the surface layer of the soil surrounding the pools (10 cm under the surface), *T. latifolia* had decomposition rate constants of 0.56 $yr^{-1}$ in the treed and 0.39 $yr^{-1}$ in the open area while *S. capillifolium* had decomposition
rate constants of 0.09 $yr^{-1}$ to 0.07 $yr^{-1}$ respectively (Table 1).

Two-way ANOVAs showed no evidence of difference in decomposition rate constants among pools ($P = 0.524$) or interacting effect of pool and depth on decomposition ($P = 0.965$) (Table S1). In all pools, there was strong evidence that decomposition rates for *T. latifolia* were faster at the surface than at the bottom of the pools and in the sediments (Figure 2; Table 1, $P <$



0.011) but no evidence of differences in decomposition rates between the bottom of the pools and in the sediments ($P = 0.978$)

(Table S2). There was little difference in decomposition rates for *S. capillifolium* between the different depths ($P = 0.1$) (Table 1; Table S3). Overall, *T. latifolia* had faster decomposition rates than *S. capillifolium* at the surface ($P < 0.001$), but not at depth ($P = 0.248$) (Table 1; Table S4). Decomposition rates seemed also faster in the surface layer of the peat than at any depth in the pools for *T. latifolia* (Table 1; $P < 0.001$) but were similar in all locations for *S. capillifolium* (Table 1; $P > 0.1$). These results suggest that both environmental conditions and litter chemistry are important drivers of OM decomposition in peatland

pools.

Table 1. Single-exponential linear regression models and percentage of remaining mass (± standard deviation) of *Typha latifolia* and *Sphagnum capillifolium* litterbags after 27 months of incubation in 6 pools of different morphometry (depth at the center and surface area shown) and in the unsaturated, surficial layer of the surrounding peat. Litterbags were incubated at the

surface of the pools (depth = 0), 15 cm over the pool bottoms in the center of the pools (depth = +15), and 15 cm below the pools, in the sediments (depth = -15). In the peat, litterbags were installed ~5 cm under the surface. The single-exponential linear regression model ($\ln[y] = a - kt$) was based on the percentage of mass remaining (y), with a = intercept, $k$ = decay rate constant, and t = time in years. % MR = percentage of original mass remaining in the litterbags after 27 months. Standard errors (± SE) are shown for parameters a, $k$ and % MR.

| Pool | Depth | *Typha latifolia* | | | | *Sphagnum capillifolium* | | | |
|---|---|---|---|---|---|---|---|---|---|
| | | a | $k$ (yr⁻¹) | r² | % MR | a | $k$ (yr⁻¹) | r² | % MR |
| G1 | 0 | 4.57 ± 0.03 | 0.19 ± 0.02 | 0.83 | 67.7 ± 1.4 | 4.59 ± 0.02 | 0.07 ± 0.01 | 0.53 | 89.3 ± 4.1 |
| (91 cm; | +15 | 4.59 ± 0.01 | 0.08 ± 0.01 | 0.83 | 84.5 ± 3.0 | 4.61 ± 0.02 | 0.07 ± 0.01 | 0.54 | 87.4 ± 0.8 |
| 190 m²) | -15 | 4.60 ± 0.02 | 0.07 ± 0.01 | 0.54 | 89.4 ± 2.6 | 4.66 ± 0.03 | 0.09 ± 0.02 | 0.54 | 90.7 ± 4.0 |
| G2 | 0 | 4.58 ± 0.03 | 0.10 ± 0.02 | 0.66 | 74.8 ± 5.5 | 4.57 ± 0.02 | 0.06 ± 0.01 | 0.53 | 90.7 ± 1.1 |
| (106 cm; | +15 | 4.57 ± 0.03 | 0.05 ± 0.02 | 0.22 | 89.3 ± 2.9 | 4.58 ± 0.01 | 0.04 ± 0.01 | 0.57 | 90.1 ± 0.7 |
| 10 m²) | -15 | 4.59 ± 0.02 | 0.07 ± 0.02 | 0.51 | 88.3 ± 2.6 | 4.62 ± 0.01 | 0.07 ± 0.01 | 0.73 | 90.2 ± 1.0 |
| G3 | 0 | 4.56 ± 0.02 | 0.16 ± 0.02 | 0.85 | 68.3 ± 4.4 | 4.58 ± 0.01 | 0.10 ± 0.01 | 0.85 | 79.8 ± 0.3 |
| (170 cm; | +15 | 4.59 ± 0.01 | 0.05 ± 0.01 | 0.68 | 87.9 ± 1.9 | 4.59 ± 0.01 | 0.04 ± 0.01 | 0.51 | 94.4 ± 0.6 |
| 180 m²) | -15 | 4.59 ± 0.02 | 0.07 ± 0.01 | 0.65 | 86.1 ± 2.4 | 4.62 ± 0.02 | 0.04 ± 0.02 | 0.29 | 96.3 ± 1.7 |
| G4 | 0 | 4.61 ± 0.02 | 0.17 ± 0.01 | 0.92 | 70.1 ± 4.2 | 4.56 ± 0.02 | 0.05 ± 0.01 | 0.42 | 87.5 ± 4.2 |
| (60 cm; 35 | +15 | 4.61 ± 0.01 | 0.08 ± 0.01 | 0.85 | 81.6 ± 1.0 | 4.61 ± 0.03 | 0.08 ± 0.02 | 0.48 | 85.7 ± 6.0 |
| m²) | -15 | 4.60 ± 0.02 | 0.07 ± 0.01 | 0.65 | 84.4 ± 9.1 | 4.62 ± 0.03 | 0.08 ± 0.02 | 0.47 | 87.0 ± 4.0 |
| G5 | 0 | 4.58 ± 0.02 | 0.16 ± 0.01 | 0.91 | 69.7 ± 4.0 | 4.57 ± 0.03 | 0.11 ± 0.02 | 0.63 | 77.1 ± 4.0 |
| (107 cm; | +15 | 4.59 ± 0.02 | 0.08 ± 0.01 | 0.69 | 83.7 ± 4.4 | 4.60 ± 0.02 | 0.05 ± 0.01 | 0.41 | 90.6 ± 0.8 |
| 129 m²) | -15 | 4.59 ± 0.03 | 0.08 ± 0.02 | 0.46 | 85.2 ± 8.0 | 4.63 ± 0.02 | 0.09 ± 0.02 | 0.60 | 81.2 ± 3.8 |
| G6 | 0 | 4.57 ± 0.02 | 0.10 ± 0.01 | 0.78 | 80.5 ± 0.6 | 4.60 ± 0.02 | 0.09 ± 0.02 | 0.66 | 81.6 ± 6.1 |
| (177 cm; | +15 | 4.58 ± 0.02 | 0.03 ± 0.01 | 0.18 | 94.5 ± 6.4 | 4.59 ± 0.01 | 0.04 ± 0.01 | 0.59 | 89.3 ± 1.5 |
| 48 m²) | -15 | 4.60 ± 0.01 | 0.07 ± 0.00 | 0.92 | 85.8 ± 1.2 | 4.64 ± 0.02 | 0.08 ± 0.02 | 0.52 | 88.7 ± 2.7 |
| Acrotelm, treed area | | 4.74 ± 0.09 | 0.58 ± 0.06 | 0.82 | 28.1 ± 6.9 | 4.56 ± 0.02 | 0.09 ± 0.02 | 0.52 | 79.5 ± 5.0 |
| Acrotelm, open area | | 4.68 ± 0.07 | 0.42 ± 0.06 | 0.82 | 38.5 ± 6.0 | 4.58 ± 0.03 | 0.07 ± 0.03 | 0.31 | 87.3 ± 5.4 |






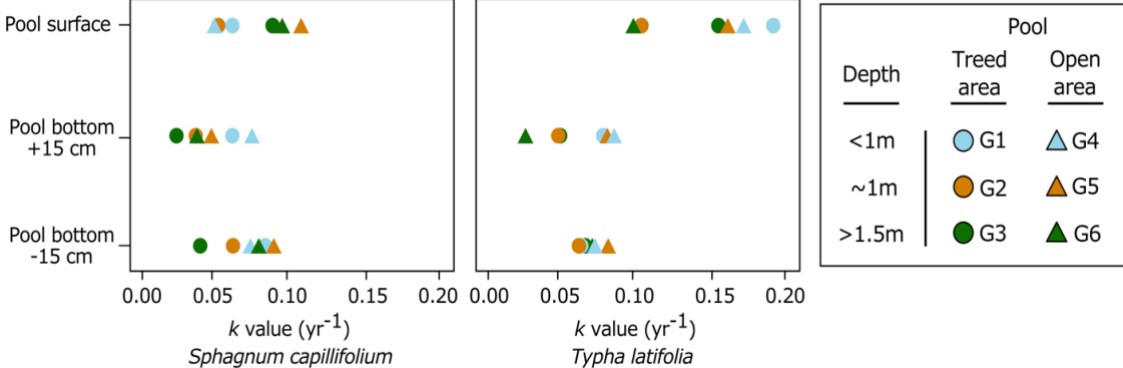

Figure 2. Decomposition rate constants (yr$^{-1}$) of *Sphagnum capillifolium* and *Typha latifolia* along the pool depth profiles in both treed and open areas of the peatland after 27 months of incubation.

### 3.1.2 Effects of incubation depth on litter chemistry

Distinctive patterns in litter chemistry appeared after 27 months of incubation. At the end of the experiment, there was no detectable change in C concentration at any position of incubation for both litter type, with an overall final C concentration of 106 ± 1 % and 98 ± 3 % that of the initial material for *T. latifolia* and *S. capillifolium*, respectively. For *T. latifolia*, there was generally a net relative gain of both N and P compared to C, with respective final concentrations of 334 ± 78 % and 192 ± 41 % that of the initial material. The highest N concentration increase was measured at the surface of the pools (416 ± 64 % N of the initial concentration vs 303 ± 61 % at the bottom and 284 ± 23 % in the sediments), but there were little variations among

depths of incubation for P (191 ± 31% of the initial concentration at the surface, 190 ± 65 % at the bottom, 195 ± 25 % in the sediments). For *S. capillifolium*, there were large variations in N and P with respective final concentrations of 102 ± 22 % and 65 ± 12 % that of the initial material, primarily indicative of over time P loss. Final N and P concentrations were generally higher at the surface of the pools (119 ± 20 % N and 70 ± 16 % P of the initial material), then at the bottom (95 ± 16 % N and

63 ± 9 % P) and in the sediments (91 ± 8 % N and 60 ± 7 %). Spearman correlations showed that C/N (ρ = 0.50, *P* = 0.002) and N/P (ρ = -0.59, *P* < 0.001) ratios in both *T. latifolia* and *S. capillifolium* respectively increased and decreased with depth of incubation, regardless of the pool (Figure 3).

We measured changes in litter isotopic signatures in relation to the position of litterbags in the pools, with values of δ$^{13}$C for *S. capillifolium* slightly increasing at depth, from -27.5 ‰ at the surface to -27.3 ‰ at the bottoms of the pools and in the

sediments. For *T. latifolia*, δ$^{13}$C also slightly increased at depth from -27.0 ‰ at the surface to -26.6 ‰ at the bottoms of the pools and to -26.7 ‰ in the sediments. Changes in δ$^{15}$N followed the same trends for both litter types, with decreasing values from surface > bottom of the pools > sediments (0.45 ‰ > -1.10 ‰ > -2.59 ‰ and -5.97 ‰ > -7.72 ‰ > -7.90 ‰ for *T. latifolia* and *S. capillifolium*, respectively).



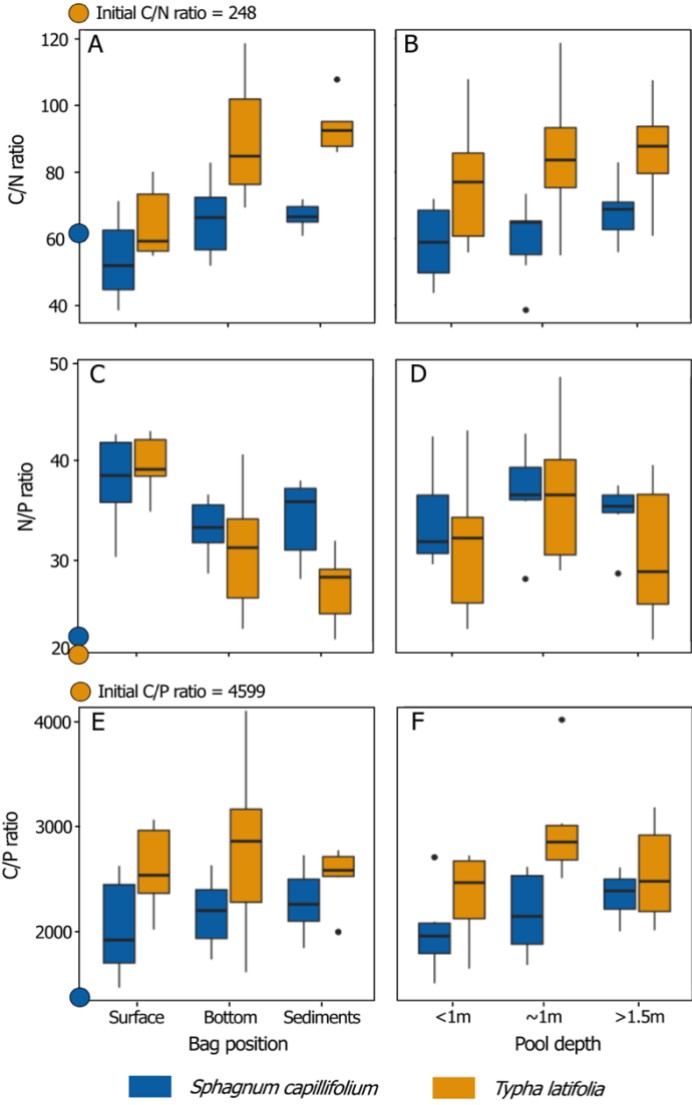

Figure 3. *Sphagnum capillifolium* (blue) and *Typha latifolia* (orange) C:N:P ratios after 27 months in relation to position of incubation in the pools (surface, at the bottom, and in the sediments, all pools together) and pool depth category (<1 m, ~ 1 m, and > 1.5 m, regardless of depth of incubation). Boxes show the median and the 25th and 75th percentiles of the distributions, and whiskers show the 10th and 90th percentiles. Colored dots on the Y-axes of the graphs show the initial chemistry of the litter.

For both litter types, there was no variation in humification indices (HIs) among position of incubations and all HI increased over time, except for the 1720/1090 ratio of *S. capillifolium* which decreased from 0.51 in the initial material to 0.47 on average at the end of the experiment (Figure S1). Compared to the initial material, HIs 1510/1090 and 1630/1090 were higher for both





*S. capillifolium* and *T. latifolia* after 27 months in the pools. There was also an increase in HI 1720/1090 (carboxylic acids and aromatic esters relative to carbohydrates) over time for *S. capillifolium*, but a decrease for *T. latifolia.*

### 285    3.1.3 Relationships between litter chemistry and decomposition rates

There were strong relationships between decomposition rates and litter chemistry for *T. latifolia*, as shown by the GLMs of decomposition rates and $\delta^{13}C$, $\delta^{15}N$ and C/N and N/P ratios, (Figure 4B-E; Table S5). There were also strong interactions between *T. latifolia* decomposition rates and humification indices (1510/1090 and 1630/1090), showing increasing proportions of aromatics and proteins relative to carbohydrates with increasing decomposition rates (Figure 4F-I). There was however no

discernible relationship between decomposition rates and OM chemistry for *S. capillifolium*.

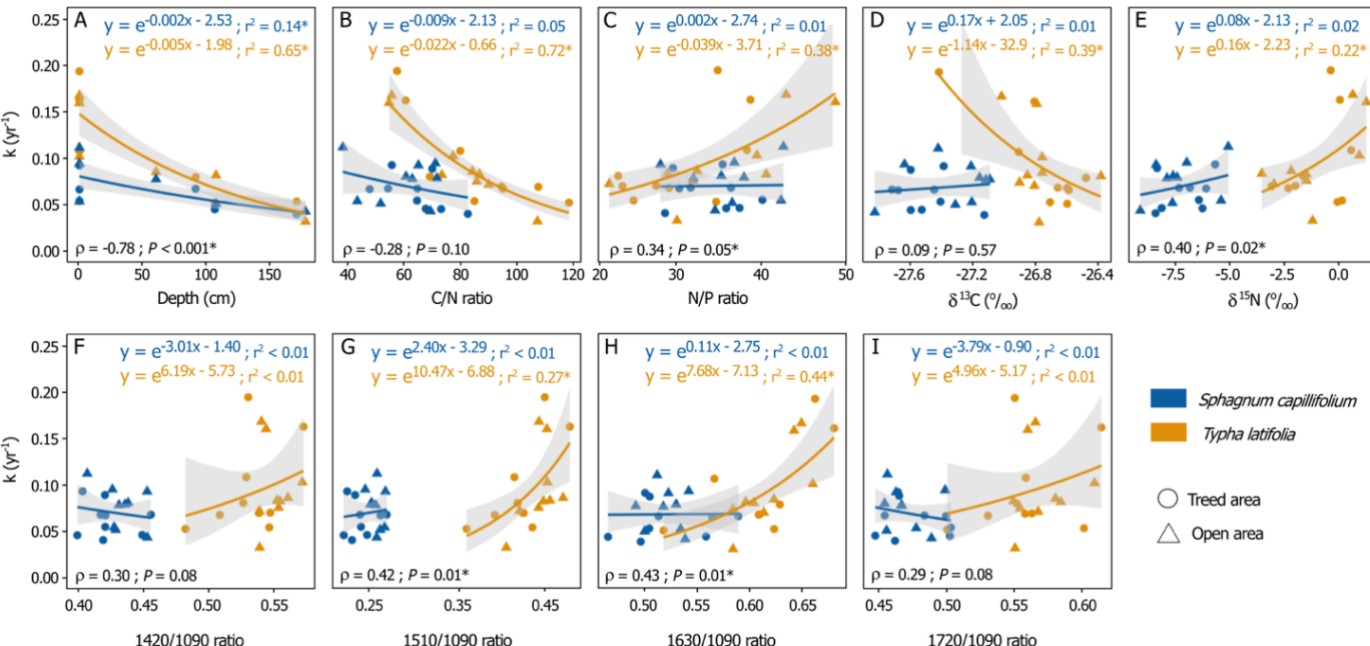

Figure 4. Average decomposition rates (*k*) and 95% confidence intervals for *Sphagnum capillifolium* and *Typha latifolia* in relation to A) depth of incubation in the pools, and final litter chemistry (after 27 months of incubation in the pools: B) C/N ratio, C) N/P ratio, D) $\delta^{13}C$ (‰), E) $\delta^{15}N$(‰), and humification indices (ratios of MIRS intensities) at F) 1420 cm$^{-1}$, G) 1510

cm$^{-1}$, H) 1630 cm$^{-1}$, and I) 1720 cm$^{-1}$ to the intensity at 1090 cm$^{-1}$. In A), bags that were incubated in the pool sediments are not shown. Colors (blue and orange) and shapes (circles and triangle) respectively represent litter type and the peatland area in which the litter was incubated (treed or open area). Generalized linear models for individual litter types using gamma distribution (log-link function), model $r^2$, and Spearman correlation ($\rho$) and p-values (*P*) of the relationships of *k* values to the identified properties, regardless of litter type, are shown. Models and correlations with p-values < 0.05 are indicated with an

asterisk (*).





Regardless of litter type, there were correlations between decomposition rates and residual (i.e., after 27 months) litter chemistry (Figure 4B-I). Decomposition rates were indeed negatively correlated to C/N ratio ($\rho = -0.28$, $P = 0.1$) and positively correlated to N/P ratio ($\rho = 0.34$, $P = 0.05$) and $\delta^{15}N$ ($\rho = 0.40$, $P = 0.02$). Similarly, litter humification indices were all

positively correlated to decomposition rates: 1420/1090 ($\rho = 0.30$, $P = 0.08$), 1510/1090 ($\rho = 0.42$, $P = 0.01$), 1630/1090 ($\rho = 0.43$, $P = 0.01$), and 1720/1090 ($\rho = 0.29$, $P = 0.08$). Regardless of environmental conditions, these results suggest close relationships between litter chemistry and decomposition rates.

### 3.1.4 Effects of pool morphology and chemistry on litter decomposition

The six pools we selected had different physical and chemical characteristics. Pools located in the treed area, though tests

showed little evidence of differences in depth to pools of the open area (Kruskal-Wallis; $P = 0.967$), generally had higher DOC concentration ($P = 0.007$), higher SUVA ($P = 0.044$) and lower pH ($P = 0.002$) (Table 2). Regardless of the differences in vegetation assemblages surrounding the pools, deep pools (> 1.5 m) had lower TN ($P = 0.002$), TP ($P = 0.009$) and $NO_3$ ($P = 0.015$) concentrations than shallow (< 1 m) and average (~ 1 m) pools, which had similar chemistry (Table 2). Individually, pools also had different chemical characteristics. For example, pool G2 had higher DOC and TP concentrations than any other

pool throughout the study (Table 2). However, GLMs showed few relationships between water chemistry and litter mass loss (Table S6), indicating that mass losses mostly vary depending on litter type and on the position in the pool (Figure 2).

### 3.2 Decomposability of pool sediments under controlled conditions

### 3.2.1 Spatial variations in sediment chemistry and $CO_2$ and $CH_4$ production

There were large differences in sediment chemistry among pools and between areas of the peatland (Table 3). For example, P

concentrations were higher in pools of the open area of the peatland while the lowest and highest N concentrations were respectively recorded in the shallowest pools of the treed and of the open area. There were also large variations in $\delta^{13}C$ and humification indices among pools.

After 35 days of incubation, mean total production for $CO_2$ and $CH_4$ respectively ranged between 1393 and 2449 µg $CO_2$ g $C^{-1}$, and -0.03 and 123 µg $CH_4$ g $C^{-1}$ (Table 3). The highest total $CO_2$ and $CH_4$ production were from ~1 m deep pools from both

the treed and open areas, while shallow and deep pools had similar production. There was however little evidence of differences in $CO_2$ (Kruskal-Wallis; $P = 0.35$) and $CH_4$ (Kruskal-Wallis; $P = 0.44$) production between pools of the treed and open areas. Production rates of $CO_2$ were usually highest in the first week of incubation and decreased afterward (Figure 5C). For all pools, rates varied over time and between pools but were generally similar among replicates. For $CH_4$, total production and production rates were close to zero in the first days of incubation but generally increased from day 5 to day 35, possibly indicating the

development of anaerobic conditions in the jars past the first week that would not be counterbalanced by the addition of



ambient air after subsampling (Figure 5C-D). There was no significant correlation between $CO_2$ and $CH_4$ production rates ($\rho$ = 0.04, $P$ = 0.75).

Table 2. Mean values ($\pm$ standard deviation) of pool water physical and chemical variables measured at the beginning and at the end of the 2019, 2020 and 2021 growing seasons. Letters in indices show levels of difference in water chemistry between individual pools for variables measured at every sampling time as determined by Kruskal-Wallis and Dunn post-hoc tests at $P$ < 0.05.

| | Treed area | | | Open area | | |
|---|---|---|---|---|---|---|
| | G1 | G2 | G3 | G4 | G5 | G6 |
| Depth (cm) | 91 ($\pm$ 4)[a] | 106 ($\pm$ 9)[b] | 170 ($\pm$ 5) [c] | 60 ($\pm$ 10) [d] | 107 ($\pm$ 5) [b] | 177 ($\pm$ 4) [c] |
| *Area (m$^2$) | 190 | 10 | 180 | 35 | 129 | 48 |
| pH | 4.12 ($\pm$ 0.20)[ab] | 3.86 ($\pm$ 0.19)[a] | 4.11 ($\pm$ 0.15)[ab] | 4.36 ($\pm$ 0.17)[b] | 4.41 ($\pm$ 0.15)[b] | 4.13 ($\pm$ 0.17)[ab] |
| DOC (mg L$^{-1}$) | 27.6 ($\pm$ 13.2)[a] | 39.6 ($\pm$ 19.9)[b] | 22.0 ($\pm$ 10.5)[a] | 21.6 ($\pm$ 10.8)[a] | 22.1 ($\pm$ 11.6)[a] | 22.6 ($\pm$ 10.8)[a] |
| TN (mg L$^{-1}$) | 1.0 ($\pm$ 0.4)[a] | 1.0 ($\pm$ 0.3)[a] | 0.6 ($\pm$ 0.1)[a] | 1.0 ($\pm$ 0.5)[a] | 1.0 ($\pm$ 0.4)[a] | 0.6 ($\pm$ 0.2)[a] |
| TP ($\mu$g L$^{-1}$) | 18.4 ($\pm$ 6.8)[a] | 51.8 ($\pm$ 41.3)[b] | 11.7 ($\pm$ 2.9)[a] | 19.6 ($\pm$ 11.3)[a] | 14.9 ($\pm$ 3.5)[a] | 10.0 ($\pm$ 3.6)[a] |
| NO$_3$ ($\mu$g L$^{-1}$) | 7.2 ($\pm$ 4.0)[ab] | 7.3 ($\pm$ 3.8)[ab] | 3.7 ($\pm$ 2.5)[a] | 6.2 ($\pm$ 5.0)[ab] | 10.9 ($\pm$ 2.4)[b] | 3.8 ($\pm$ 3.2)[a] |
| NH4 ($\mu$g L$^{-1}$) | 207.6 ($\pm$ 287.8)[a] | 47.3 ($\pm$ 23.1)[a] | 40.7 ($\pm$ 17.7)[a] | 161.3 ($\pm$ 185.5)[a] | 296.4 ($\pm$ 216.0)[a] | 34.0 ($\pm$ 15.3)[a] |
| PO$_4$ ($\mu$g L$^{-1}$) | 4.2 ($\pm$ 0.8)[a] | 5.3 ($\pm$ 2.4)[a] | 6.0 ($\pm$ 5.3)[a] | 4.3 ($\pm$ 3.4)[a] | 3.4 ($\pm$ 2.3)[a] | 3.6 ($\pm$ 2.8)[a] |
| A$_{254}$ | 1.15 ($\pm$ 0.19)[ab] | 1.67 ($\pm$ 0.41)[b] | 0.97 ($\pm$ 0.19)[a] | 0.87 ($\pm$ 0.43)[a] | 0.79 ($\pm$ 0.36)[a] | 0.87 ($\pm$ 0.13)[a] |
| SUVA$_{254}$ (L mg C$^{-1}$ m$^{-1}$) | 4.17 ($\pm$ 0.31)[a] | 4.28 ($\pm$ 0.24)[a] | 4.31 ($\pm$ 0.26)[a] | 4.02 ($\pm$ 1.05)[a] | 3.80 ($\pm$ 1.26)[a] | 3.93 ($\pm$ 0.42)[a] |
| [†]Dissolved CO$_2$ (mg L$^{-1}$) | 1.24 | 7.85 | 1.85 | 2.28 | 1.50 | 1.86 |
| [†]Dissolved CH$_4$ (mg L$^{-1}$) | 0.061 | 1.44 | 0.036 | 0.40 | 0.17 | 0.011 |
| [†]Dissolved N$_2$O ($\mu$g L$^{-1}$) | 0.9 | 0.6 | 0.9 | 0.8 | 0.9 | 0.9 |
| [†]Flux CO$_2$ (mg CO$_2$ m$^{-2}$ hr$^{-1}$) | 42.5 | 208.2 | 100.9 | 148.7 | 111.8 | 83.8 |
| [†]Flux CH$_4$ (mg CH$_4$ m$^{-2}$ hr$^{-1}$) | 0.76 | 13.32 | 0.54 | 9.61 | 4.68 | 0.15 |

*Pool area was measured once using a 2020 0.12m x 0.12m resolution satellite photo. [†]GHG dissolved concentrations and fluxes were only measured at the end of the 2021 growing season.

### 3.2.2 Effects of sediment chemistry on $CO_2$ and $CH_4$ production

Production of $CO_2$ and $CH_4$ varied between pools and replicates in relationship to different sediment chemical properties (Figure 6). Both $CO_2$ and $CH_4$ production tended to be higher for sediments with high P content (Table 3). Humification indices were also negatively correlated to $CO_2$ production (1420:1090 ratio, $\rho$ = -0.37, $P$ = 0.05; 1630:1090 ratio, $\rho$ = -0.31, $P$ = 0.09; 1720:1090 ratio, $\rho$ = -0.36, $P$ = 0.05) but had no effect on $CH_4$ ($P$ > 0.1) (Table 4). For $CH_4$, sediment C content ($\rho$ = 0.41, $P$




= 0.02), $\delta^{13}C$ ($\rho$ = -0.34, $P$ = 0.07) and N:P ratio ($\rho$ = -0.35, $P$ = 0.06) had an influence on production (Table 4). Content in Na and K were also negatively correlated to $CH_4$ but not to $CO_2$ production (Table 4).

Table 3. Mean total production for $CO_2$ and $CH_4$ and chemical properties of pool sediments ($\pm$ standard deviation, n = 5 replicates per pool) after 35-day incubations. For gas production, negative values are considered consumption. Letters in indices show levels of difference between individual pools as determined by Kruskal-Wallis and Dunn post-hoc tests at p-values < 0.05.

| | Treed area | | | Open area | | |
|---|---|---|---|---|---|---|
| | G1 | G2 | G3 | G4 | G5 | G6 |
| Depth (cm) | 91 ($\pm$ 4)[a] | 106 ($\pm$ 9)[b] | 170 ($\pm$ 5) [c] | 60 ($\pm$ 10) [d] | 107 ($\pm$ 5) [b] | 177 ($\pm$ 4) [c] |
| $CO_2$ production ($\mu$g $CO_2$ g $C^{-1}$) | 1393 ($\pm$ 581)[a] | 2324 ($\pm$ 940)[a] | 1415 ($\pm$ 218)[a] | 1573 ($\pm$ 579)[a] | 2449 ($\pm$ 731)[a] | 1720 ($\pm$ 598)[a] |
| $CH_4$ production ($\mu$g $CH_4$ g $C^{-1}$) | -0.03 ($\pm$ 0.02)[a] | 123 ($\pm$ 222)[a] | 0.2 ($\pm$ 0.3)[a] | -0.6 ($\pm$ 1.6)[a] | 3.0 ($\pm$ 6.3)[a] | 10.2 ($\pm$ 11.5)[a] |
| C (g $g^{-1}$) | 0.506 ($\pm$ 0.012)[ab] | 0.517 ($\pm$ 0.002)[c] | 0.508 ($\pm$ 0.003)[abc] | 0.501 ($\pm$ 0.003)[b] | 0.501 ($\pm$ 0.003)[b] | 0.513 ($\pm$ 0.003)[ac] |
| N (g $g^{-1}$) | 0.033 ($\pm$ 0.001)[a] | 0.029 ($\pm$ 0.001)[b] | 0.035 ($\pm$ 0.001)[c] | 0.034 ($\pm$ 0.001)[a] | 0.041 ($\pm$ 0.000)[d] | 0.035 ($\pm$ 0.000)[c] |
| P ($\mu$g $g^{-1}$) | 299 ($\pm$ 4)[a] | 409 ($\pm$ 15)[b] | 255 ($\pm$ 8)[c] | 369 ($\pm$ 11)[d] | 448 ($\pm$ 17)[e] | 518 ($\pm$ 13)[f] |
| C/N ratio | 15.4 ($\pm$ 0.2)[a] | 17.7 ($\pm$ 0.4)[b] | 14.4 ($\pm$ 0.3)[c] | 14.9 ($\pm$ 0.5)[ac] | 12.3 ($\pm$ 0.1)[d] | 14.6 ($\pm$ 0.3)[c] |
| C/P ratio | 1689 ($\pm$ 55)[ab] | 1265($\pm$ 43)[acd] | 1990 ($\pm$ 71)[b] | 1358 ($\pm$ 43)[abc] | 1119 ($\pm$ 35)[cd] | 992 ($\pm$ 22)[d] |
| N/P ratio | 110 ($\pm$ 3)[a] | 72 ($\pm$ 2)[b] | 138 ($\pm$ 3)[c] | 91 ($\pm$ 1)[d] | 91 ($\pm$ 3)[d] | 68 ($\pm$ 2)[b] |
| $\delta^{13}C$ (‰) | -27.26 ($\pm$ 0.01)[a] | -27.73 ($\pm$ 0.05)[b] | -27.43 ($\pm$ 0.02)[c] | -25.61 ($\pm$ 0.10)[d] | -26.03 ($\pm$ 0.05)[e] | -28.67 ($\pm$ 0.06)[f] |
| $\delta^{15}N$ (‰) | -1.75 ($\pm$ 0.02)[a] | -1.89 ($\pm$ 0.05)[ab] | -1.57 ($\pm$ 0.04)[c] | -1.98 ($\pm$ 0.07)[b] | -2.03 ($\pm$ 0.07)[b] | -1.91 ($\pm$ 0.14)[ab] |
| 1420/1090 ratio | 0.641 ($\pm$ 0.010)[ab] | 0.668 ($\pm$ 0.017)[bc] | 0.689 ($\pm$ 0.018)[c] | 0.628 ($\pm$ 0.012)[a] | 0.616 ($\pm$ 0.009)[a] | 0.663 ($\pm$ 0.021)[bc] |
| 1510/1090 ratio | 0.687 ($\pm$ 0.013)[a] | 0.697 ($\pm$ 0.020)[ab] | 0.738 ($\pm$ 0.017)[c] | 0.675 ($\pm$ 0.019)[a] | 0.720 ($\pm$ 0.014)[bc] | 0.756 ($\pm$ 0.019)[c] |
| 1630/1090 ratio | 1.188 ($\pm$ 0.020)[abc] | 1.173 ($\pm$ 0.033)[ab] | 1.246 ($\pm$ 0.015)[d] | 1.155 ($\pm$ 0.017)[ab] | 1.179 ($\pm$ 0.018)[ab] | 1.218 ($\pm$ 0.030)[cd] |
| 1720/1090 ratio | 0.791 ($\pm$ 0.014)[ac] | 0.818 ($\pm$ 0.018)[b] | 0.822 ($\pm$ 0.027)[b] | 0.763 ($\pm$ 0.023)[c] | 0.718 ($\pm$ 0.024)[d] | 0.777 ($\pm$ 0.027)[ac] |





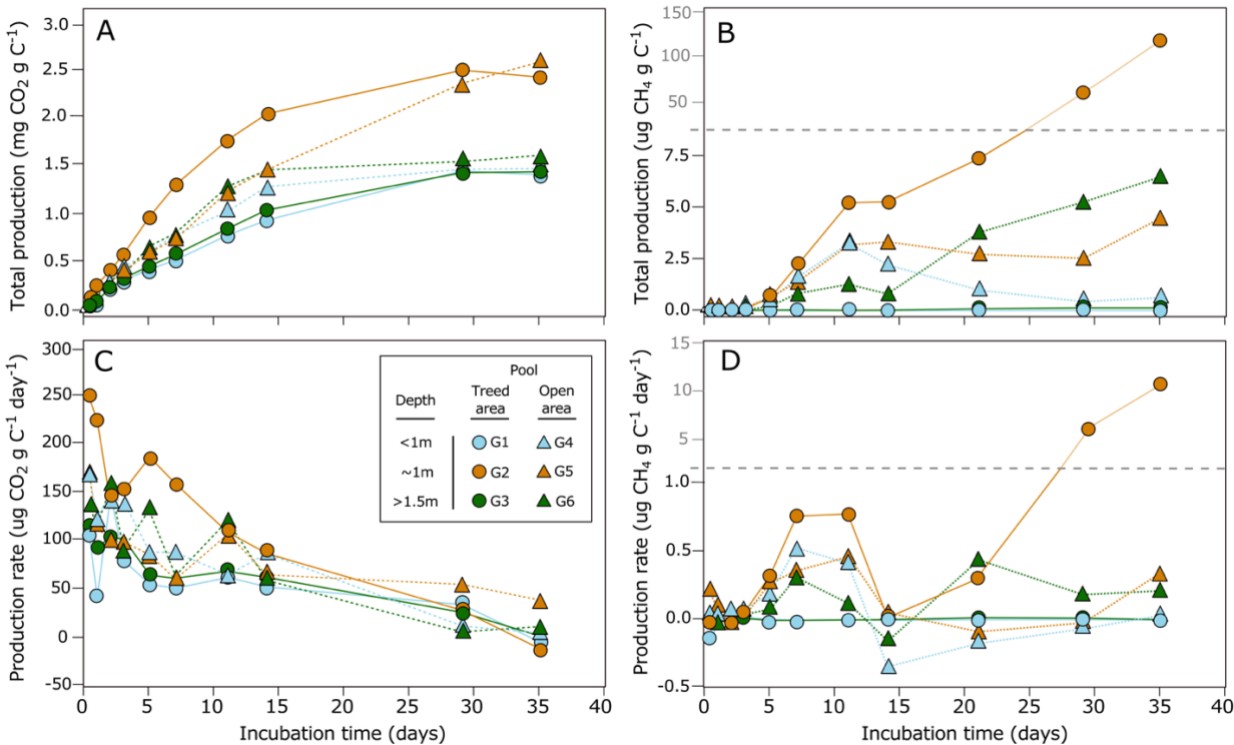

Figure 5. A) Total $CO_2$ and B) total $CH_4$ produced from the incubation of the first 30 cm of bottom material (including limnic material and the underlying peat) of the six studied pools (G1 to G5) over the 35-day period. C) Potential production rates for $CO_2$ (µg $CO_2$ g peat$^{-1}$ day$^{-1}$) and D) $CH_4$ (µg $CH_4$ g peat$^{-1}$ day$^{-1}$) for each pool sediment. Dots are average values of five replicates.

### 3.3 Comparing OM decomposition rates from fresh litter and pool sediments

Spatial patterns emerged when comparing fresh OM decomposition rates at the bottom of each pool and $CO_2$ and $CH_4$ production from pool sediments (Figure 7). $CO_2$ ($P = 0.43$) and $CH_4$ ($P = 0.35$) production rates were not significantly different between the shallowest and deepest pools, but decomposition rates of *S. capillifolium* and *T. latifolia* were ($P = 0.03$). Opposingly, $CO_2$ production was higher in sediments from pools that were ~1 m deep than in other pools ($P = 0.01$), but decomposition rates of fresh litter were not significantly different ($P = 0.20$). Therefore, there seemed to be little relationship between $CO_2$ and $CH_4$ production rates from pool sediments and litter decomposition rates.





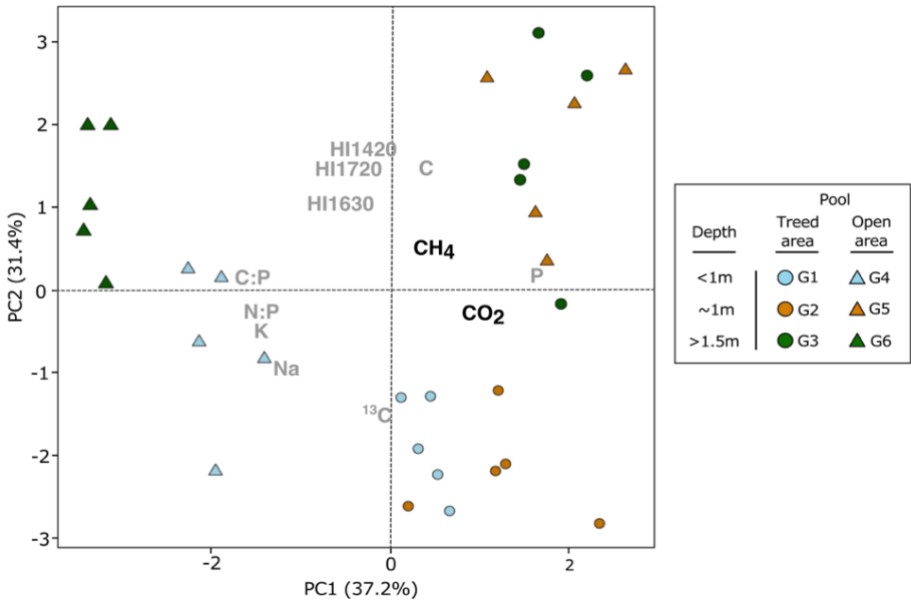

Figure 6. Principal component analysis of pool sediment chemistry and carbon dioxide ($CO_2$) and methane ($CH_4$) total production after 35 days of incubation. Only chemical variables that were correlated to $CO_2$ and $CH_4$ total production were included in the analysis (Table 4; $P < 0.1$).

Table 4. Spearman's rank correlation coefficient ($\rho$) of $CO_2$ and $CH_4$ total production and sediment chemical properties. Asterisks show correlations with p-values ($P$) < 0.05 (**) or < 0.1 (*).

| | CO₂ production | | CH₄ production | |
|---|---|---|---|---|
| | $\rho$ | $P$ | $\rho$ | $P$ |
| C content | 0.01 | 0.98 | 0.41 | 0.02 ** |
| N content | 0.13 | 0.49 | -0.11 | 0.55 |
| P content | 0.32 | 0.08 * | 0.27 | 0.15 |
| C:N ratio | -0.11 | 0.55 | 0.13 | 0.50 |
| N:P ratio | -0.23 | 0.23 | -0.35 | 0.06 * |
| C:P ratio | -0.33 | 0.07 * | -0.26 | 0.16 |
| $\delta^{13}C$ | -0.05 | 0.78 | -0.34 | 0.07 * |
| $\delta^{15}N$ | -0.27 | 0.14 | 0.19 | 0.30 |
| 1420:1090 ratio | -0.37 | 0.05 ** | 0.24 | 0.21 |
| 1510:1090 ratio | -0.16 | 0.39 | 0.14 | 0.45 |
| 1630:1090 ratio | -0.32 | 0.09 ** | 0.14 | 0.45 |
| 1720:1090 ratio | -0.36 | 0.05 ** | 0.18 | 0.33 |
| Ca content | -0.14 | 0.43 | -0.11 | 0.55 |
| K content | -0.21 | 0.25 | -0.32 | 0.08 * |
| Mg content | -0.05 | 0.77 | -0.27 | 0.15 |
| Na content | -0.20 | 0.28 | -0.49 | 0.01 ** |



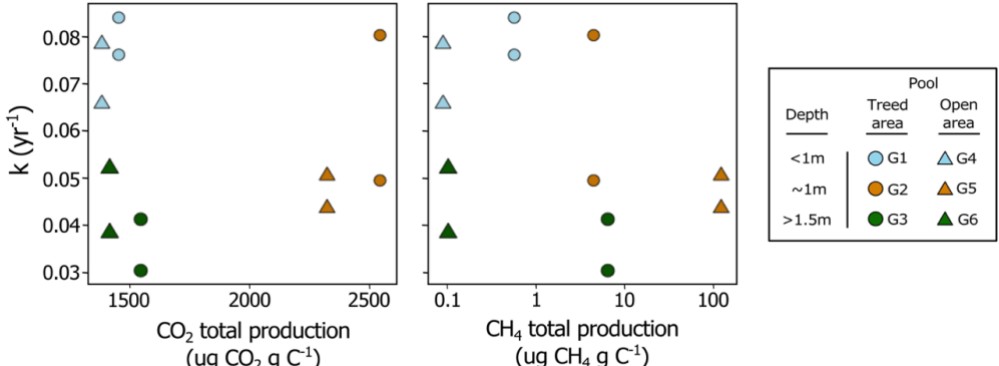

Figure 7. Relationship between fresh litter decomposition rates ($k$) at the bottom of each pool and carbon dioxide ($CO_2$) and methane ($CH_4$) total production from pool sediments after 35 days of incubation. On each graph, there are two identical symbols per pool, one for *Sphagnum capillifolium* and one for *Typha latifolia*.

## 4 Discussion

### 4.1 Decomposability of fresh OM in peatland pools

Studies on peatland pools are currently lacking to fully understand the drivers of the decomposition processes that occur within pools and, more generally, their biogeochemical patterns and our study is a step towards filling this gap in the literature. Our results show variations in decomposition rates of standardized litter among pools, among depths of incubation and also between the litter types, with decomposition rates for *T. latifolia* being much higher than for *S. capillifolium*. Differences in environmental conditions, biological activity and litter quality are major drivers of spatial variability in OM decomposition in terrestrial and aquatic systems (Bradford et al., 2016; Joly et al., 2023). For example, decomposition rates of *Sphagnum* species are usually slow compared to that of vascular plants like *T. latifolia* because of their high content in recalcitrant organic compounds (e.g. phenolics and sphagnan) and low nutrient concentrations (Hájek et al., 2011). In our study, the depth at which fresh OM decomposition occurred was the most important driver of spatial variations in decomposition rates. At the surface of the pools, decomposition rates for *S. capillifolium* and *T. latifolia* were faster than at any other depth and, at the bottom, slowed with increasing pool depth. In aquatic ecosystems, litter decomposition rates are positively correlated to $O_2$ availability and temperature (e.g. Hardwick et al., 2022; Lecerf et al., 2007) and are faster under high light intensity (Kuehn et al., 2014) because of increased microbial activity. At the study site, summer dissolved $O_2$ concentrations, water temperature and light intensity all decreased with increasing depth from the pool surface (Arsenault et al., 2018). Therefore, we argue that the differences in the decomposition rates we measured were largely driven by within-pool variations in environmental conditions





(i.e. $O_2$ concentration, temperature, and light) that could influence the structure of microbial communities and the magnitude of decomposition processes.

Dissolved $O_2$ concentration is a major control of litter decay in freshwater systems (LaRowe and Van Cappellen, 2011). When $O_2$ concentrations decrease, other biogeochemical processes intensify (e.g. anaerobic respiration or fermentation) and this results in slower decomposition rates (Blodau, 2011). Dissolved $O_2$ in aquatic systems is positively correlated to litter decomposition rates because the proportion of microorganisms with the ability to decompose litter in microbial communities is larger under high $O_2$ availability (Liu et al., 2022). This relationship may then explain the slower decomposition rates for

both *S. capillifolium* and *T. latifolia* with increasing depth from pool surface (Figure 4A), as peatland pool water is depleted in $O_2$ at depth. Thus, decomposition of any type of litter material that falls in deep pools would be slower than in shallow pools. Given the spatial heterogeneity in pool depth that exists in many temperate peatlands (Arsenault et al., 2023), the effects pools have on the C balance of the peatlands in which they develop may then be larger than currently expected.

Warmer temperature increases decomposition rates in both aquatic and terrestrial environments (e.g. Kim et al., 2015; Treat et

al., 2014). A mesocosm experiment has shown that warmer water temperature accelerated the decomposition of plant material in aquatic systems and increased the activity and diversity of decomposition-related bacteria (Pan et al., 2021). In our study, water temperature at the pool surface was similar in all pools suggesting other environmental drivers were controlling the differences in litter decomposition among pools at the surface. However, during the growing season when most of the biological activity occurs, pool water temperature decreased with increasing depth from the surface (Arsenault et al., 2018),

supporting the idea that spatial variations in litter decomposition within pools were partly related to water temperature. With increasing summer temperature under current and future climate warming, litter decomposition rates may increase in peatland pools, especially in the shallowest ones, and considerably affect their C budget.

Light has an indirect effect on litter decomposition as it influences microbial biomass and activity (Danger et al., 2013; Halvorson et al., 2019). For example, in aquatic ecosystems, light may increase algae growth and the production of labile OM

(Singh and Singh, 2015) that could stimulate the overall microbial activity of decomposition (Halvorson et al., 2019). Increased light intensity at the surface than in depth could then enhance litter decomposition in the upper layer of the water column. Light penetration is however influenced by dissolved and suspended matter (Wetzel, 2001). In our pools, we estimated dissolved matter concentration and composition by measuring water color ($A_{254}$) and SUVA. Water color was negatively correlated to decomposition rates of *T. latifolia* at the surface and of both litter types in depth, meaning that light attenuation

due to dissolved matter may influence litter decomposition rates in peatland pools. However, we did not measure suspended matter in our pools which could possibly have explained differences in litter decomposition rates among pools and areas (Table 1). Without identifying light as a clear driver of C cycling in pools, our results nonetheless indicate that possible pool water browning in response to climate change (Fenner et al., 2021) may counterbalance some of the effects the expected higher temperatures during the growing season may have on litter decomposition.





Decomposition rates for both *S. capillifolium* and *T. latifolia* in the pool sediments were similar to those measured at the bottom of the pools meaning that the environmental conditions 15 cm below the pools were not as limiting as we might have expected, even though we measured hypoxia below the water-sediment interface (Arsenault et al., 2018). In peatlands or in aquatic ecosystems, litter decomposition rates are slower in anoxic than in oxic conditions (e.g. Moore et al., 2007; Neckles & Neill, 1994). Liu et al. (2022) however found that litter decomposition was still effective in anaerobic conditions because of distinct

structures in microorganism communities. It is then possible that similar variations in microbial communities sustained fresh litter decomposition in the anoxic, low temperature and light-free conditions that normally prevail in peatland pool sediments.

**4.2 Decomposability of pool sediments**

The results of our incubation experiment showed that OM chemistry could only partly explain spatial variations in pool sediment decomposition. For example, $CO_2$ production was positively related to an increase in OM humification, but $CH_4$

production was mostly related to C content but not composition (Table 4). Hence, while the incubations were conducted under the same conditions and samples were treated in the same way, our results do not point towards a specific control of inherent sediment properties on decomposition rates. In fact, the relatively small influence sediment chemistry exerts on its degradability is supported by the results of the litterbag component of the study. Indeed, we observed relationships between OM chemistry and decomposition for *T. latifolia* but not for *S. capillifolium* (Figure 4). Given that peat at the study site is

mostly composed of *Sphagnum* species, variations in $CH_4$ and $CO_2$ production from our pool sediments may thus be explained by other factors we did not measure, such as the structure of microbial communities found in the sediments (Oloo et al., 2016), or the effects of chemical properties were too small compared to the observed variation in the data. This emphasizes the lack of available information on peatland pools and calls for further studies on the specific mechanisms responsible for sediment degradation in such unique freshwater systems.

**4.3 Drivers of OM decomposition rates in peatland pools**

Many studies have quantified the C budget of peatlands (e.g. Yu, 2012), but little effort has been put on understanding the effect C emitting features like pools have on this budget, or on determining the factors controlling C emissions from such features. Here, we argue that OM decomposition rates in peatland pools are controlled by several mechanisms that are regulated by the environmental condition in which this decomposition occurs. For example, both litterbag and incubation experiments

indicate controls of OM decomposition in peatland pools that are not only chemical, but also physical (e.g. pool morphology) and biological (e.g. surrounding vegetation composition). Spatial variations in fresh litter decay and pool sediment decomposition did not follow the same trends, and there was no relationship between both components of the study (Figure 7). While we may relate $CO_2$ and $CH_4$ production to OM chemistry, litter decomposition in pool water primarily slowed with increasing depth from the pool surface. This suggests that the chemistry of OM alone could not explain variations in OM

decomposition in peatland pools.



Overall, $CO_2$ and $CH_4$ production were highest in the ~1 m deep pools (pools G2 and G5; Table 3, Figure 5A), in which OM chemistry may be ideal to support high rates of decomposition. For example, along with pool G6 which also had high $CO_2$ and $CH_4$ production (Figure 5), pools G2 and G5 had the largest sediment P concentration (Table 3). Given that P is limiting in peatland pools (Arsenault et al., 2018), higher P concentrations in the substrate may enhance its decomposition (Moore et al., 2008). The negative correlation between $CO_2$ production and humification indices of the remaining litter after incubation (Table 4) is also indicative of chemical controls of sediment decomposability, with higher $CO_2$ production from sediments that contained higher relative proportions of polysaccharides to proteins and aromatics. The relationships of $CO_2$ production to humification indices are in agreement with the hypothesis of reduced decomposition rates with increasing OM aromaticity. Because we would expect the sediments of shallower pools to be composed of less degraded peat, higher $CO_2$ production from the ~1 m deep pool and the correlations between $CO_2$ and humification indices also show that the chemical composition of pool sediments and peat vary spatially but that potentially other mechanisms we did not study, e.g. related to the structure of microbial communities, are key drivers of OM decomposition in peatland pools. Then, we suggest that pool depth, by influencing the environmental, and possibly biological and thermodynamic conditions in which decomposition occurs, is a more important control of decomposition processes in peatland pools than OM chemistry. Peatland pools are often overlooked (Loisel et al., 2017), hence the mechanisms leading to $CO_2$ and $CH_4$ production in pools, the relationship between such production and pool structure, and, more generally, the role pools play in the C budget of peatlands are unclear. Overall, our results indicate that the degradation potential of peatland pool OM is spatially heterogeneous, with large variations emerging in fresh litter and sediment decomposition within a small area of a temperate ombrotrophic peatland. This means that the estimation of the C budget of a given peatland with pools may not only be imprecise, but also that current knowledge on pool biogeochemistry and decomposition processes may not be readily upscaled and transposable to pools of different geographic settings.

Peatland pool biogeochemistry responds strongly to changes in temperature (Arsenault et al., 2023). A recent study in a subboreal raised bog has also shown that pool surface area, hence possibly also depth, have declined in the last six decades because of increased mean air temperature (Colson et al., 2023). With our results clearly showing that environmental conditions are a major driver of OM decomposition, this means that decomposition rates in peatland pools may be increasing over time in response to decreasing pool depth under climate change. Indeed, regardless of litter type, we have measured the highest rates of decomposition in the warmest and more oxygenated locations of our pools. Then, expected structural changes in peatland pools under climate change (e.g. Karofeld & Tõnisson, 2014) would likely lead to changes in the magnitude of their C emission function, but this still needs to be verified.

## 5 Conclusion


Our study showed that OM decomposition in peatland pools is highly variable and depends primarily on the environmental conditions in which it occurs, with spatial patterns emerging in both fresh litter and pool sediment decomposability as a function of pool depth (Figure 7). This means that, regardless of the type of litter that falls at the bottom of the pools, we may expect different decomposition rates in pools of different depths in response to distinct biogeochemical mechanisms and

pathways. With depth being the main driver of variations in OM decomposition, as it controls dissolved $O_2$ concentrations, light penetration and water temperature at the pool bottom, it is thus possible that changes in peatland hydrology and expected warmer summer temperature under climate change will modify both the structure and the biogeochemical function of peatland pools.

### Data availability

The data that support the findings of this study are openly available on Zenodo at https://doi.org/10.5281/zenodo.10581234.

### Author contribution

JA, JT, JFL and TRM designed the study. JA carried the litterbags and incubation experiments. HT performed the litter and peat chemical analyses. JA analysed the data and prepared the manuscript with contributions from all co-authors.

### Competing interests

The authors declare that they have no conflict of interest.

### Acknowledgments

We thank all students from the Laboratoire d'Étude des Fonctions Écosystémiques, de leur Stabilité Spatiotemporelle et des Enjeux Socioenvironnementaux at UdeM who have provided priceless help in installing and retrieving the experimental setup at GPB. We also thank Mike Dalva for his help during the incubation phase of the research. This research was funded by

NSERC and FRQNT through PhD fellowships to JA, and research funding from NSERC Discovery grant to JT (RGPIN-2020-05310).

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
