# Peer review of "Patterns and drivers of organic matter decomposition in peatland open-water pools"

_EGUsphere, 2024_

## Referee Comment (RC1)

**Review of: Patterns and drivers of organic matter decomposition in peatland open-water pools**

Julien Arsenault, Julie Talbot, Tim R. Moore, Klaus-Holger Knorr, Henning Teickner, Jean- François Lapierre

**General Comments:**
The paper outlines an interesting study of peatland pools in Canada. They combine a litterbag decomposition experiment with lab-based incubations to understand the drivers of OM decomposition and GHG emissions from peat sediments. Peatland pools are under-studied – this paper is a significant contribution to research on these interesting environments. The justification for the research is clear, and the methods are well described. The results are sometimes a bit unclear, but there are some really interesting results hidden in there. The implications for C emissions and climate need to be more explicit.

**Abstract:**
As it is a complex experiment, there is a lot of information in the abstract, which makes it a bit more difficult to read. However, it generally outlines the research well.

**Introduction:**
The introduction outlines existing research briefly and succinctly. There are a few papers on temperate peatland pools that missing from this summary of past research (relating to pools in the UK Forsinard Flows, and small ditches in Sweden) that could be included to support biogeochemistry findings (especially re: depth, DOC and dissolved $CO_2$ in pools).

**Methods:**
Figure 1 is good, it's a clear and concise explanation of the experimental design.
*Typha latifolia* isn't found at the site, why did you choose it for the litter bags? Was it because it isn't found there naturally? It would be good to include a justification for this.
Do the measurements are line 104-105 mean that you did a site survey of 158 of the 600 pools? Why were the litterbags collected multiple times during the experiment? The paper focusses in those collected at the end of the experiment – did the ones collected earlier not show any interesting results?
At line 130, do you mean 'for up to 27 months' here? Or were molecular composition analyses only done on litter from the final sampling occasion?
What does the 't' stand for at line 172?

**Results:**
3.1.1. Figure 2 and Table 1 show similar results – does Table 1 add anything to the story that isn't covered by Figure 2? The values for intercept, %MR and r2 aren't discussed in the text, maybe they could go in SI?
3.1.2. At line 255, you say there is a distinctive pattern, and then say there is no detectable change – and then say there was 106% and 98% (which I would say is a change, even if it's a small one). This gives mixed signals.
The results in this section are very interesting.
Line 267: 'regardless of the pool' – does this mean pool depth, or pool number/location?

The caption of Figure 3 was difficult to interpret, where you refer to pool depth category, and then have 'regardless of depth of incubation'. I see why you have used that wording, but is there a way to make it clearer which 'depth' is which?

In Figure 3, can you include the number of samples that each box represents? (e.g. n=2)

Line 280: HI – can you remind what the different HI ratios mean here?

When you say 'increased over time', do you mean between initial and final weight, or for all litterbag retrievals between the beginning and end?

3.1.3. Are the results in Figure 4A the same as those in Figure 2? The k values at different depths? Figure 4 shows a lot of information very clearly.

3.1.4. The second sentence in this section is hard to understand. Can you re-word it?

3.2.1. The sentence at line 320 makes it sound like the P concentrations were higher in the pool rather than the sediments.

The lowest value of $CH_4$ production in Table 3 is -0.6, but in the sentence at line 323, you write the range as between '-0.03 and 123 ug $CH_4$'

Figure 5 caption – '...of the six studied pools (G1 to G5)...' what about pool G6? Also please check you are referring to the correct graphs/gases in the text (especially at line 331). I don't think you refer to Figure 5A in the results text at all.

3.2.2. No attempt to explain the results or PCA here.

3.3. You write that 'spatial patterns emerged' but then end this section saying 'there seemed to be little relationship between $CO_2$ and $CH_4$ production rates....' – so what are the spatial patterns?

**Discussion:**
4.1. The sentence at line 407 needs a bit more explaining please.

4.2. This section is clear and concise.

4.3. The header of this section sounds the same as section 4.1.

In section 4.1. you state that $O_2$ concentration, light and temperature are drivers of litter decomposition in peat pools, whereas in this section you talk about P content of the sediment. The result at line 465 is very interesting and wasn't mentioned earlier in the study (or wasn't highlighted as much as it could be).

**Supplementary information:**
Tables S1-S4 could be condensed into one table, just showing p values and post-hoc test results for each test? I don't think that knowing the degrees of freedom, sum of squares and mean square of each test adds much.

Figure S1 repeats the C/N and N/P ratio graphs that are in Figure 3 – probably unnecessary.

**Text edits with line numbers:**
Line 15: The sentence is long and contains a lot of information, could it be rewritten as two sentences to make it clearer?

Line 71: hence and hence and hence – too many hences.

Line 224: '... was more degraded than...'

---

## Author Comment (AC1)

**Referee # 2**

Our responses are in red in the text below.

The study assesses spatial variation in organic matter (OM) decomposition rates in peatland pools at a temperate peatland site. The authors also evaluated the influence of OM chemistry on decomposition rates of fresh litter in situ and on sediments ex situ. The authors' finding that decomposition rates vary spatially in peatland pools is highly relevant for assessments of peatland greenhouse gas (GHG) emissions and the paper will be of interest to the readership of Biogeosciences.

Thank you for the thoughtful and constructive review. Here we provide a point by point answer to the concerns you have raised.

The research approach and methods are well-designed overall, however the authors should justify the decision to oven-dry litter samples for the in situ decomposition experiment, and compare decomposition rates of litter to other studies in the discussion.

The litters were oven dried at 40°C to ensure that a uniform residual moisture content existed, particularly for *Sphagnum* which is difficult to dry uniformly and that desiccation would discourage *Sphagnum* from regrowing. We will specify this in the revised ms.

The comparison with decomposition rates of other litters and locations is valid as they are the same species and are treated in the same way. Regarding comparison with other studies, we will address this in the revised ms. We present below some comparisons as an answer to another question raised by the referee.

The threshold for statistical significance should be clarified in the methods.

The significance level we used is $P < 0.05$ (except in Table 4, where we used either $P < 0.05$ or $P < 0.1$ but this is specified). However, we refrained from using the terms 'significant' in most of the text (except in three occasions where P-values were $> 0.2$) and rather chose to give the reader every $P$-value and, ultimately, the opportunity to interpret the significance of each statistic. We therefore believe it is not necessary to clarify the significance level in the methods section.

The paper presents a comprehensive dataset and the interpretation and discussion of results needs to be re-examined and strengthened in some instances. A few issues stand out:

According to the introduction OM decomposition is generally faster in pools than in surrounding soils, but in this study k values tended to be much higher in the acrotelm compared to the pools

for T. latifolia and k values appear to be similar for pools and the acrotelm in the case of S. capillifolium (Table 1). This should be discussed.

Here, we are not totally sure what the referee refers to. In the introduction, we say that pools 'generally have a negative C balance because of distinct environmental conditions (e.g. higher oxygen – $O_2$ – availability and warmer temperature) compared to the surrounding soils that lead to faster OM decomposition than production'. We do not imply that decomposition is faster in the pools than in the soil, but rather that decomposition occurs faster than accumulation. To avoid any confusion, we will rephrase to: "[...] that lead to rates of OM decomposition that are faster than rates of OM production decomposition than production".

We also believe the differences in decomposition rates for *Typha* and *Sphagnum* in the acrotelm vs. in the pools reinforce our justification to use both an easily degradable (*Typha*) and a more recalcitrant substrate (*Sphagnum*) to distinguish the effects of environmental and chemical properties on litter decomposition rates.

While the authors propose that decomposition rate in pools depends primarily on environmental conditions (i.e. depth), there was no difference in decomposition rates of S. capillifolium among depths (Table 1). It seems there are dual influences of environment and litter composition on decomposition rates *in situ* that the authors should consider more carefully.

Yes, there are environmental and litter quality influences at play, and a core objective of our study was to disentangle to what extent each aspect plays a role. We found that where the litter decomposition rate is 'relatively' fast (*Typha*), environment does play a role in differentiating rates. Where it is 'slow' over the short period (27 months; i.e. *Sphagnum*) patterns are not clear though they may have appeared if mass loss was larger.

The slow rates of decomposition of *S. capillifolium* in the open-pools are similar to those found in the surface of bog and fen peatlands in temperate and boreal regions (e.g. Moore and Basiliko 2006; Moore et al. 2007) and there is little differentiation of rates within the pools. The faster rates of decomposition of *T. latifolia* at the surface of the open-pools (0.10 to 0.19) is similar to rates obtained in the surface of peatlands and in freshwater marshes, where k values range from 0.14 to 0.59 (e.g. Moore et al. 2007; Dong et al. 2024). The decrease in the *k* value with depth of emplacement in the open-pools is also consistent with studies that placed the *T. latifolia* leaves at depth in peatlands, resulting in a strong correlation between the *k* value and the period of saturation derived from the water table position, declining from k values of about 0.2 where never saturated to 0.05 when continuously saturated (Moore et al. 2007; Clarkson et al. 2014). This is consistent with the open-pool decline in *k* value from an average of 0.15 near the water surface to an average of 0.07 in the pool sediment, suggesting that anaerobic conditions are a major control on decomposition rates of fresh organic matter in these peatlands. We will add these numbers and references to the discussion in the revised ms to better support our hypotheses.

Dong, D., P. Badiou, T.R. Moore, C. von Sperber 2024. Litter decomposition and nutrient dynamics of four macrophytes in intact and restored freshwater marshes of Canada. *Restoration Ecology* 32 issue 4. doi.org/10.1111/rec.14135.

Moore, T.R., J.L. Bubier and L.A. Bledzki 2007. Litter decomposition in temperate peatlands: the effect of substrate and site. *Ecosystems* 10: 949-963.

Moore, T. and N. Basiliko 2006. Decomposition. Pages 126-143 in R.K. Wieder & D.H. Vitt (eds.) *Boreal Peatland Ecosystems*, Ecological Studies Vol. 188, Springer-Verlag.

Clarkson, B.R., T.R. Moore, N.B. Fitzgerald, D. Thornburrow, C.H. Watts and S. Miller 2014. Water table regime regulates litter decomposition in restiad peatlands, New Zealand. *Ecosystems* 17: 317-326. doi:10.1007/s10021-013-9726-4.

Regarding the influence of OM chemistry on decomposition of sediments ex situ statements in the abstract and discussion contradict each other.  The abstract states that CO2 production by sediments decreases with increasing OM humification and at line 440 it is stated that "CO2 production was positively related to an increase in OM humification". The abstract states that CH4 production decreases with increasing N:P but this result is not significant at $p < 0.05$ (Table 4), though there is a significant correlation between CH4 and C content as well as Na content at $p < 0.05$ (see comment above about clarification of threshold for determination of statistical significance). This section of the discussion doesn't address the potential mechanisms explaining these correlations, though an explicit aim of the study is to "assess the role OM chemistry plays on decomposition rates for litter and sediments" and an implied aim is to increase knowledge of mechanisms controlling OM decomposition in peatland pools.

Thank you for pointing to what we stated at line 440. This is clearly a typo, as shown by the results in Table 4. We will correct this mistake in the revised ms.

We agree that there are inconsistencies between our discussion and how we present results in the abstract and how to identify potential drivers of decomposition processes in peatland pools. To address this, we believe we should better highlight what we identify as correlations in the discussion (plus limitations due to small sample sizes) and, more importantly, that we only make suggestions for what could be causes for the observed relations. We will then remove less clear relations from the abstract.

It is also true that one of our main objectives is not entirely met. We will address this in the revised ms and discuss why we did not meet this objective by emphasizing the potential mechanisms that could explain the correlations our results point to by citing relevant literature. For example:

- The importance of peat quality in controlling OM decomposition, as suggested by the negative correlation between $CO_2$ production and humification indices and previously observed by Estop-Aragonés et al. (2022).

- A possible P limitation in our systems (as suggested by the positive correlations between C emissions and P concentration in the material; Figure 6, Table 4) rather than N limitation as observed in permafrost settings (Schädel et al. 2014).

Estop-Aragonés, C., L. Heffernan, K.H. Knorr, D. Olefeldt 2022 Limited Potential for Mineralization of Permafrost Peatland Soil Carbon Following Thermokarst: Evidence From Anoxic Incubation and Priming Experiments *JGR Biogeosciences* 127: e2022JG006910.

Schädel, C., E.A.G. Schuur, R. Bracho, B. Elberling, C. Klobauch, H. Lee, Y. Luo, G.R. Shaver, M.R. Turetsky 2014. Circumpolar assessment of permafrost C quality and its vulnerability over time using long-term incubation data *Global Change Biology* 20: 641-642.

The abstract states that differences in fresh litter and pool sediment decomposability is a function of O2 concentrations, light, and temperature, which all decrease with increasing depth. Variation in these environmental parameters with depth are not presented in the present study but it seems they were included in a previous paper (Aresenault et al. 2018). The relationship between decomposition rates and O2, light, and temperature should be presented in the present study to substantiate this conclusion.

This is an interesting suggestion that could greatly improve the message we are sending in the paper. Unfortunately, we only measured such parameters on 6 occasions during the 27-months period of our field work (at t=0 and then at every litter retrieving time, to be precise). Hence, we do not have a large enough sample size on which to base correlations. We must then rely on other studies, one of which (Arsenault et al. 2018) was conducted at the same study site but on different pools. We will specify this in the discussion in the revised ms and remove the mention of these specific drivers in the abstract because we did not directly measure them..

It's also not clear how sediment decomposition rate can vary with depth since the sediments were collected from the bottoms of the pools. Perhaps the authors mean overall pool depth, but the ex situ decomposition rates were highest at the intermediate pool depth, not the shallowest pools. It's interesting that the total ex situ production of CO2 corresponds to the sediments from the pools with the highest CO2 fluxes in situ. The authors have focused on the influence of physical factors but it seems that chemistry is also important.

We indeed meant general pool depth, this will be clarified in the revised ms.

Our incubation study shows that decomposability (or decomposition potential) is indeed higher in the intermediate pool depth group (~1 m deep) than elsewhere, but our chemical analyses of individual sediment pool samples do not point to many correlations that could explain such results (see Table 4). We also believe that sediment chemistry is important in influencing sediment decomposition, but our experiment being the first to directly study decomposition processes in peatland pools, it is thus difficult to put forward hypotheses without falling into simple speculations.

This is why we call, at line 448 of the submitted ms, "for further studies on the specific mechanisms responsible for sediment degradation". To avoid any misunderstanding, we will also emphasize on the hypothetical nature of the interpretation we make from the results presented in Figures 5 to 7 in the revised ms.

---

## Author Comment (AC2)

**Referee #1**

Our responses are in red in the text below.

**General Comments:**

The paper outlines an interesting study of peatland pools in Canada. They combine a litterbag decomposition experiment with lab-based incubations to understand the drivers of OM decomposition and GHG emissions from peat sediments. Peatland pools are under-studied – this paper is a significant contribution to research on these interesting environments. The justification for the research is clear, and the methods are well described. The results are sometimes a bit unclear, but there are some really interesting results hidden in there. The implications for C emissions and climate need to be more explicit.

Thank you for the thoughtful and constructive review. Below we provide a point by point answer to the concerns raised by the Reviewer, including better expliciting the implications for C emissions and climate.

**Abstract:**

As it is a complex experiment, there is a lot of information in the abstract, which makes it a bit more difficult to read. However, it generally outlines the research well.

The abstract is indeed dense because our article combines two complementary experiments. We nonetheless agree that it can be difficult to read and will be restructuring the abstract to separate the results of both components of the study more clearly.

**Introduction:**

The introduction outlines existing research briefly and succinctly. There are a few papers on temperate peatland pools that missing from this summary of past research (relating to pools in the UK Forsinard Flows, and small ditches in Sweden) that could be included to support biogeochemistry findings (especially re: depth, DOC and dissolved $CO_2$ in pools).

We are aware that several studies conducted in the UK and Sweden have looked at peatland water bodies. However, we omitted to refer to most of them because they were conducted in different environmental settings (e.g. disturbed or restored peatlands, blanket bogs) than our undisturbed raised bog. Previous studies have shown that differences in environmental conditions influence the biogeochemistry of peatland pools (especially C cycling) and thus the decomposition processes occurring within (e.g. Chapman et al. 2022; Arsenault et al. 2023). We therefore focused the introduction on peatland pools of similar context.

We will address this comparison issue in the introduction to better contextualize our research settings.

Chapman, P. J., C.S. Moody, T.E. Turner, R. McKenzie, K.J. Dinsmore, A.J. Baird, M.F. Billett, R. Andersen, F. Leith and J. Holden 2022. Carbon concentrations in natural and restoration pools in blanket peatlands. *Hydrological Processes* 36:e14520.

Arsenault, J., J. Talbot, L.E. Brown, M. Helbig, J. Holden, J. Hoyos-Santillan, E. Jolin, R. Mackenzie, K. Martinez-Cruz, A. Sepulveda-Jauregui and J.F. Lapierre 2023. Climate-driven spatial and temporal patterns in peatland pool biogeochemistry. *Global Change Biology* 29: 4056-4068.

**Methods:**

Figure 1 is good, it's a clear and concise explanation of the experimental design.

Thank you.

*Typha latifolia* isn't found at the site, why did you choose it for the litter bags? Was it because it isn't found there naturally? It would be good to include a justification for this.

*Typha latifolia* was chosen for several reasons, that we will synthesize in the methodology section of the paper:

It can be cut into consistent, standard lengths and breadths and it is easy to remove material brought into the litterbags during extraction (it can be tedious on softer materials).

It decomposes moderately quickly (for wetland vegetation) so we were assured of there being a reasonable mass loss during the duration, *cf* Sphagnum which we knew would decompose slowly and may show no significant differences with emplacement location.

One author has used *Typha latifolia* from the same source in a variety of wetland locations as a 'standard' litter, providing an assessment of the 'quality' of the environment for decomposition. Thus, we can compare the 'decomposability potential of the environment' against, for example, Canadian bogs and fens, New Zealand peatlands and wetlands in Ontario and Manitoba:

Moore, T.R., J.L. Bubier and L.A. Bledzki 2007. Litter decomposition in temperate peatlands: the effect of substrate and site. *Ecosystems* 10: 949-963.

Clarkson, B.R., T.R. Moore, N.B. Fitzgerald, D. Thornburrow, C.H. Watts and S. Miller 2014. Water table regime regulates litter decomposition in restiad peatlands, New Zealand. *Ecosystems* 17: 317-326.

Dong, D., P. Badiou, T.R. Moore, C. von Sperber 2024. Litter decomposition and nutrient dynamics of four macrophytes in intact and restored freshwater marshes of Canada. *Restoration Ecology*.

Do the measurements are line 104-105 mean that you did a site survey of 158 of the 600 pools?

Yes. Since 2015, we indeed have measured 158 of the ~600 pools found at the site. Temporal surveys showed little changes in water-level (only several centimeters over the course of a growing season; Arsenault et al., 2019) and satellite photos show no discernible changes in pool extent over time. We are then confident that both depth and area did not change much since the first pools were measured. To avoid any further question on the number of pools we surveyed, we will specify this in the methodology section of the paper.

Arsenault, J., J. Talbot, T.R. Moore, M.P. Beauvais, J. Franssen and N.T. Roulet 2019. The spatial heterogeneity of vegetation, hydrology and water chemistry in a peatland with open-water pools. *Ecosystems* 22: 1352-1367.

Why were the litterbags collected multiple times during the experiment? The paper focusses in those collected at the end of the experiment – did the ones collected earlier not show any interesting results?

The litterbags were collected several times to provide an indication of the times during the year when decomposition occurs and to provide data to estimate the exponential decay parameters. We therefore used data (litter weight loss after X months) from each of the five times we collected litterbags. We will specify this in the last paragraph of section 2.2.

At line 130, do you mean 'for up to 27 months' here? Or were molecular composition analyses only done on litter from the final sampling occasion?

As stated, only the initial samples and those exposed for 27 months were analyzed. We proceeded that way because we were mostly interested in how litter changed after three summers in the pools rather than the temporal evolution of litter chemistry. The in-between samples (4, 11, 16 and 23 months after installation) were used to better estimate mass loss parameters, but not for quality changes. We will rewrite parts of the last paragraph of section 2.2 to avoid any confusion.

What does the 't' stand for at line 172?

At line 172, 't' stands for 'tons'. We will specify it in the text.

**Results:**

3.1.1. Figure 2 and Table 1 show similar results – does Table 1 add anything to the story that isn't covered by Figure 2? The values for intercept, %MR and r2 aren't discussed in the text, maybe they could go in SI?

We have also pondered whether we kept it in the main text or SI, and decided that although Table 1 would probably fit well in Supplementary information, we believe it provides the background to the rather simple Figure 2, and the variability as shown by the standard error, which would complicate Figure 2.

3.1.2. At line 255, you say there is a distinctive pattern, and then say there is no detectable change – and then say there was 106% and 98% (which I would say is a change, even if it's a small one). This gives mixed signals.

There is a distinctive change in the overall litter chemistry, but we feel that the range of C concentration at the end of the decomposition period was between 98 and 106% of that initial concentration is not worthy of comment, compared to the changes in N and P. We will clarify this in the revised version of the paper.

The results in this section are very interesting.

Thank you.

Line 267: 'regardless of the pool' – does this mean pool depth, or pool number/location?

We meant 'pool number/location'.  To avoid any confusion, we will rephrase to : "In all six pools, sSpearman correlations showed that C/N ($\rho = 0.50$, $P = 0.002$) and N/P ($\rho = -0.59$, $P < 0.001$) ratios in both *T. latifolia* and *S. capillifolium* respectively increased and decreased with depth of incubation."

The caption of Figure 3 was difficult to interpret, where you refer to pool depth category, and then have 'regardless of depth of incubation'. I see why you have used that wording, but is there a way to make it clearer which 'depth' is which?

We will rephrase in the revised ms to clarify what we mean, by changing "regardless of depth of incubation" to "regardless of position of incubation within the pools"

In Figure 3, can you include the number of samples that each box represents? (e.g. n=2)

We will include the number of samples in the revised ms.

Line 280: HI – can you remind what the different HI ratios mean here?

We will do as suggested.

When you say 'increased over time', do you mean between initial and final weight, or for all litterbag retrievals between the beginning and end?

We mean differences between the initial litter chemistry and that of the material after 27 months as we did not analyze the chemistry of the litter at the other retrieving times. This should be clearer after clarifying the earlier comment on the confusion among incubation times.

3.1.3. Are the results in Figure 4A the same as those in Figure 2? The k values at different depths? Figure 4 shows a lot of information very clearly.

Thank you. Yes, the k values are indeed the same for both figures, but the messages are different: in Figure 2 we wanted to clearly differentiate decomposition rates among pools and incubation positions while in Figure 4A, we wanted to show the statistical relationship between incubation depth and k values. We therefore believe both figures are relevant and should be kept.

3.1.4. The second sentence in this section is hard to understand. Can you re-word it?

We will rephrase to: "Pools located in the treed area generally had higher DOC concentration ($P = 0.007$), higher SUVA ($P = 0.044$) and lower pH ($P = 0.002$) than pools of the open area (Table 2), although tests showed little evidence of differences in pool depth between both areas (Kruskal-Wallis; $P = 0.967$)" in the revised ms to clarify what we mean.

3.2.1. The sentence at line 320 makes it sound like the P concentrations were higher in the pool rather than the sediments.

This sentence is indeed confusing. We will rephrase in the revised ms to clarify that we are specifically talking about sediment chemistry, not pool water.

3.2.2. No attempt to explain the results or PCA here.

We will describe and explain more thoroughly the results presented in this section, especially those shown by the PCA, in the revised ms.

The lowest value of $CH_4$ production in Table 3 is -0.6, but in the sentence at line 323, you write the range as between '-0.03 and 123 ug $CH_4$'

Indeed. This is a typo that will be corrected in the revised ms.

3.3. You write that 'spatial patterns emerged' but then end this section saying 'there seemed to be little relationship between $CO_2$ and $CH_4$ production rates....' – so what are the spatial patterns?

We meant that pools of different morphology and location cluster differently when comparing $CH_4/CO_2$ production from decomposing sediments and k values of decomposing fresh litter (Fig. 7). But when comparing all pairs of variables (k vs $CO_2$ and k vs $CH_4$, regardless of pool morphology and location), there was indeed little correlation between k and GHG production. We agree that this paragraph is confusing and we will rephrase in the revised ms to clarify what we mean.

Figure 5 caption – '…of the six studied pools (G1 to G5)…' what about pool G6? Also please check you are referring to the correct graphs/gases in the text (especially at line 331). I don't think you refer to Figure 5A in the results text at all.

There is indeed a typo in the Figure 5 caption that will be corrected in the revised ms. We will also carefully check that we refer to the correct figures and results in the text.

**Discussion:**

4.1. The sentence at line 407 needs a bit more explaining please.

We agree with this suggestion and will develop our hypothesis that "the effects pools have on the C balance of the peatlands in which they develop may then be larger than currently expected" in the revised ms.

This is also a great opportunity to explicitly discuss the implications our findings have on our comprehension of pools' role in peatland C emissions and how climate change may affect those, as suggested by the referee in his general comment.

4.2. This section is clear and concise.

Thank you.

4.3. The header of this section sounds the same as section 4.1.

We respectfully disagree with this comment and do not think that the headers sound the same. We first (4.1) discuss controls of fresh litter decomposition, then (4.2) we discuss controls on pool sediment decomposition, and finally (4.3) we discuss what may drive decomposition processes in peatland pools, based on what was presented in 4.1 and 4.2.

In section 4.1. you state that $O_2$ concentration, light and temperature are drivers of litter decomposition in peat pools, whereas in this section you talk about P content of the sediment.

In section 4.1, we indeed discuss the influence of dissolved oxygen, light and temperature on fresh litter decomposition processes and, in Section 4.3, we point to potential chemical and physical drivers of both fresh litter and sediments decomposition in peatland pools.

The result at line 465 is very interesting and wasn't mentioned earlier in the study (or wasn't highlighted as much as it could be).

Thank you for this comment. We will highlight this finding in the revised ms.

**Supplementary information:**

Tables S1-S4 could be condensed into one table, just showing p values and post-hoc test results for each test? I don't think that knowing the degrees of freedom, sum of squares and mean square of each test adds much.

Figure S1 repeats the C/N and N/P ratio graphs that are in Figure 3 – probably unnecessary.

We respectfully disagree with the referee's comment on the content of Supplementary information. While the tables and Figure S1 may not add much to the paper itself, we believe that some readers may find those interesting like we would. Given that they are in SI, we prefer to keep that as is.

**Text edits with line numbers:**

Line 15: The sentence is long and contains a lot of information, could it be rewritten as two sentences to make it clearer?

To answer a previous comment made by this referee about the abstract, we will restructure the abstract to separate the results of both components of the study more clearly. This way, long sentences will be shorter.

Line 71: hence and hence and hence – too many hences.

There were indeed several 'hences'. We will be rephrasing these sentences.

Line 224: '… was more degraded than…'

We will change 'degradable' to 'degraded'.

---

## Author Response (AR1)

**Response to referees**

Dear editor,

We are here submitting a revised version of our manuscript entitled 'Patterns and drivers of organic matter decomposition in peatland open-water pools'. We made several changes to the article in response to the comments and suggestions made by the two anonymous referees that reviewed our work.

We thank both anonymous referees and the editor for their thoughtful comments and suggestions to improve our paper. Our responses are listed below, in red. Line numbers are from the 'track-change' version of the ms.

In addition to responding to all comments, we also restructured parts of the discussion and conclusion to better emphasize the scope of our paper and how our findings influence our comprehension of peatland pool structure and function.

**Referee #1**

**General Comments:**

The paper outlines an interesting study of peatland pools in Canada. They combine a litterbag decomposition experiment with lab-based incubations to understand the drivers of OM decomposition and GHG emissions from peat sediments. Peatland pools are under-studied – this paper is a significant contribution to research on these interesting environments. The justification for the research is clear, and the methods are well described. The results are sometimes a bit unclear, but there are some really interesting results hidden in there. The implications for C emissions and climate need to be more explicit.

Thank you for the thoughtful and constructive review. Below we provide a point-by-point answer to the concerns raised by the Reviewer, including better expliciting the implications for C emissions and climate.

**Abstract:**

As it is a complex experiment, there is a lot of information in the abstract, which makes it a bit more difficult to read. However, it generally outlines the research well.

The abstract is indeed dense because our article combines two complementary experiments. We nonetheless agree that it can be difficult to read and restructured the abstract to separate the results of both components of the study more clearly:

At lines 15-20, the abstract now reads: "We first quantified rates of OM decomposition from fresh litter at different depths in six pools of distinct morphological characteristics in a temperate ombrotrophic peatland using litterbags of *Sphagnum capillifolium* and *Typha latifolia* over a 27-

month period. Rates of decomposition were faster for *T. latifolia* than *S. capillifolium* and, overall, faster at the pool surface and decreased with increasing depth. We then measured potential $CO_2$ and $CH_4$ production from the sediments of the same six pools by performing 35-days laboratory incubations."

We also reworded the closing statement of the abstract (lines 24-29) to summarize the implications of our findings in the context of C emissions and climate.

**Introduction:**

The introduction outlines existing research briefly and succinctly. There are a few papers on temperate peatland pools that missing from this summary of past research (relating to pools in the UK Forsinard Flows, and small ditches in Sweden) that could be included to support biogeochemistry findings (especially re: depth, DOC and dissolved $CO_2$ in pools).

We are aware that several studies conducted in the UK and Sweden have looked at peatland water bodies. However, we omitted to refer to most of them because they were conducted in different environmental settings (e.g. disturbed or restored peatlands, blanket bogs) than our undisturbed raised bog. Previous studies have shown that differences in environmental conditions influence the biogeochemistry of peatland pools (especially C cycling) and thus the decomposition processes occurring within (e.g. Chapman et al. 2022; Arsenault et al. 2023). We therefore focused the introduction on peatland pools of similar context. We addressed this comparison issue in the first paragraph of the introduction to better contextualize our research settings, at lines 39-42:

"However, C cycling in peatland pools (and thus decomposition processes occurring within) are strongly influenced by the state of the peatland (disturbed or unperturbed) and, more broadly, its geographical settings (e.g. Arsenault et al., 2023; Chapman et al., 2022). Hence, undisturbed ombrotrophic peatlands that have pool covers greater than 37% may be net sources of C to the atmosphere (Pelletier et al., 2014)."

**Methods:**

Figure 1 is good, it's a clear and concise explanation of the experimental design.

Thank you.

*Typha latifolia* isn't found at the site, why did you choose it for the litter bags? Was it because it isn't found there naturally? It would be good to include a justification for this.

*Typha latifolia* was chosen for several reasons that we now present in the methodology section of the paper (lines 121-126):

"We chose *T. latifolia* even though it was not found at the site because it decomposes moderately quickly so we were assured of measuring a reasonable mass loss during the duration of the experiment, contrary to *S. capillifolium* which decomposes slowly and may not show significant differences with litterbags emplacement location (Moore et al., 2007) *T. latifolia* from the same

source was also used in a variety of wetland locations as a standard litter, thus allowing for the comparison of the decomposability potential of peatland pools to other wetland environments (Clarkson et al., 2014; Dong et al., 2024)."

Do the measurements are line 104-105 mean that you did a site survey of 158 of the 600 pools?

There was typo in the text. Since 2015, we have measured 156 (not 158) of the ~600 pools found at the site. Temporal surveys showed little changes in water-level (only several centimeters over the course of a growing season; Arsenault et al., 2019) and satellite photos show no discernible changes in pool extent over time. We are then confident that both depth and area did not change much since the first pools were measured. To avoid any further question on the number of pools we surveyed, we now specify at lines 111-113:

"The southern and central parts of the peatland are characterized by the presence of > 600 pools of different shape, area (10-2350 $m^2$, mean 381 $m^2$; n = 156) and depth (15-219 cm, mean 93 cm; n = 156), and for which surface extent and water level vary little over time (Arsenault et al. 2018; 2019)."

Why were the litterbags collected multiple times during the experiment? The paper focusses in those collected at the end of the experiment – did the ones collected earlier not show any interesting results?

The litterbags were collected several times to provide an indication of the times during the year when decomposition occurs and to provide data to estimate the exponential decay parameters. We therefore used data (litter weight loss after X months) from each of the five times we collected litterbags.

Lines 136-138 now reads: "Litterbags were retrieved five times over the course of three growing seasons to provide an indication of the times during the year when decomposition occurs and data (litter weight loss over time) to estimate the exponential decay parameters."

At line 130, do you mean 'for up to 27 months' here? Or were molecular composition analyses only done on litter from the final sampling occasion?

As stated, only the initial samples and those exposed for 27 months were analyzed. We proceeded that way because we were mostly interested in how litter changed after three summers in the pools rather than the temporal evolution of litter chemistry. The in-between samples (4, 11, 16 and 23 months after installation) were used to better estimate mass loss parameters, but not for quality changes.

Lines 143-144 now reads: "For the samples that were retrieved after 27 months, we then milled the litter and analyzed for molecular composition of C compounds and C, N and P concentrations to assess changes between the original material and material that has been exposed to environmental conditions for 27 months in the pools."

What does the 't' stand for at line 172?

At (now) line 187, 't' stands for 'tons'. We specified it in the text.

**Results:**

3.1.1. Figure 2 and Table 1 show similar results – does Table 1 add anything to the story that isn't covered by Figure 2? The values for intercept, %MR and r2 aren't discussed in the text, maybe they could go in SI?

We have also pondered whether we kept it in the main text or SI, and decided that although Table 1 would probably fit well in Supplementary information, we believe it provides the background to the rather simple Figure 2, and the variability as shown by the standard error, which would complicate Figure 2.

3.1.2. At line 255, you say there is a distinctive pattern, and then say there is no detectable change – and then say there was 106% and 98% (which I would say is a change, even if it's a small one). This gives mixed signals.

There is a distinctive change in the overall litter chemistry, but we feel that the range of C concentration at the end of the decomposition period (between 98 and 106% of that initial concentration) is not worthy of comment, compared to the changes in N and P. Sentences at lines 269-273 are now rephrased for clarification.

The results in this section are very interesting.

Thank you.

Line 267: 'regardless of the pool' – does this mean pool depth, or pool number/location?

We meant 'pool number/location'. To avoid any confusion, we rephrased to : "In all six pools, Spearman correlations showed that C/N ($\rho = 0.50$, P = 0.002) and N/P ($\rho = -0.59$, P < 0.001) ratios in both *T. latifolia* and *S. capillifolium* respectively increased and decreased with depth of incubation." Lines 280-282.

The caption of Figure 3 was difficult to interpret, where you refer to pool depth category, and then have 'regardless of depth of incubation'. I see why you have used that wording, but is there a way to make it clearer which 'depth' is which?

We rephrased this sentence to clarify what we mean, by changing "regardless of depth of incubation" to "regardless of position of incubation within the pools and area of the peatland". Line 291.

In Figure 3, can you include the number of samples that each box represents? (e.g. n=2)

All boxes are built with six samples (n = 6). Now specified at line 292-293.

Line 280: HI – can you remind what the different HI ratios mean here?

We are now specifying what those HI ratios are at lines 296-302.

When you say 'increased over time', do you mean between initial and final weight, or for all litterbag retrievals between the beginning and end?

We mean differences between the initial litter chemistry and that of the material after 27 months as we did not analyze the chemistry of the litter at the other retrieving times. This should be clearer after clarifying the earlier comment on the confusion among incubation times.

3.1.3. Are the results in Figure 4A the same as those in Figure 2? The k values at different depths? Figure 4 shows a lot of information very clearly.

Thank you. Yes, the $k$ values are indeed the same for both figures, but the messages are different: in Figure 2 we wanted to clearly differentiate decomposition rates among pools and incubation positions while in Figure 4A, we wanted to show the statistical relationship between incubation depth and k values. We therefore believe both figures are relevant and should be kept.

3.1.4. The second sentence in this section is hard to understand. Can you re-word it?

At lines 327-329, we rephrased this sentence to: "Pools located in the treed area generally had higher DOC concentration ($P = 0.007$), higher SUVA ($P = 0.044$) and lower pH ($P = 0.002$) than pools of the open area (Table 2), although tests showed little evidence of differences in pool depth between both areas (Kruskal-Wallis; $P = 0.967$)."

3.2.1. The sentence at line 320 makes it sound like the P concentrations were higher in the pool rather than the sediments.

This sentence is indeed confusing. We rephrased it to clarify that we are specifically talking about sediment chemistry, not pool water. Line 341.

3.2.2. No attempt to explain the results or PCA here.

We rephrased the first two sentences of section 3.2.2 to imply that the PCA is indeed showing variations in CH4 and CO2 production among pools and replicates. Sentences now read: "Production of $CO_2$ and $CH_4$ varied between pools and replicates in relationship to different sediment chemical properties. For example, both $CO_2$ and $CH_4$ production tended to be higher for sediments with high P content (Figure 6; Table 4)." Lines 363-365.

The lowest value of $CH_4$ production in Table 3 is -0.6, but in the sentence at line 323, you write the range as between '-0.03 and 123 ug $CH_4$'

Indeed. This is a typo that we corrected in the text at line 345.

3.3. You write that 'spatial patterns emerged' but then end this section saying 'there seemed to be little relationship between $CO_2$ and $CH_4$ production rates....' – so what are the spatial patterns?

We meant that pools of different morphology and location cluster differently when comparing $CH_4/CO_2$ production from decomposing sediments and k values of decomposing fresh litter (Fig. 7). But when comparing all pairs of variables (k vs $CO_2$ and k vs $CH_4$, regardless of pool morphology and location), there was indeed little correlation between k and GHG production. We rephrased the first and last sentences of the paragraph to clarify what we meant. Lines 382 and 387-388.

Figure 5 caption – '…of the six studied pools (G1 to G5)…' what about pool G6? Also please check you are referring to the correct graphs/gases in the text (especially at line 331). I don't think you refer to Figure 5A in the results text at all.

There is indeed a typo in the Figure 5 caption that we corrected in the revised ms. We also carefully checked that we referred to the correct figures and results in the text.

**Discussion:**

4.1. The sentence at line 407 needs a bit more explaining please.

We agree with this suggestion and developed our hypothesis that "the effects pools have on the C balance of the peatlands in which they develop may then be larger than currently expected" in the revised ms.

At lines 435-441, we added: "Indeed, small and shallow pools emit more C than larger and deeper pools (McEnroe et al., 2009; Pelletier et al., 2014), but there is currently no inventory of the number, distribution or morphology of temperate peatland pools which limits our comprehension of their role in peatland C emissions. Then, the C balance of an ombrotrophic raised peatland with pools that are mostly small and shallow could be less than for a peatland of similar geographical settings but with pools that are mostly large and deep. This also highlights a still unknown but possible effect climate change could have on peatland C balance if peatland pool number, extent or morphology was to increase or decrease under changing environmental conditions."

4.2. This section is clear and concise.

Thank you.

4.3. The header of this section sounds the same as section 4.1.

We respectfully disagree with this comment and do not think that the headers sound the same. We first (4.1) discuss controls of fresh litter decomposition, then (4.2) we discuss controls on pool sediment decomposition, and finally (4.3) we discuss what may drive decomposition processes in peatland pools, based on what was presented in 4.1 and 4.2.

In section 4.1. you state that $O_2$ concentration, light and temperature are drivers of litter decomposition in peat pools, whereas in this section you talk about P content of the sediment.

In section 4.1, we indeed discuss the influence of dissolved oxygen, light and temperature on fresh litter decomposition processes and, in Section 4.3, we point to potential chemical and physical drivers of both fresh litter and sediments decomposition in peatland pools.

The result at line 465 is very interesting and wasn't mentioned earlier in the study (or wasn't highlighted as much as it could be).

Thank you for this comment. We added a sentence following this one to highlight this finding (lines 522-524) "This means that a possible shift in peatland vegetation composition under climate change to more shrub-like than *Sphagnum* species, as suggested by previous studies (e.g. Dieleman et al., 2015), would also influence decomposition rates and increase C emissions from peatland pools."

**Supplementary information:**

Tables S1-S4 could be condensed into one table, just showing p values and post-hoc test results for each test? I don't think that knowing the degrees of freedom, sum of squares and mean square of each test adds much.

Figure S1 repeats the C/N and N/P ratio graphs that are in Figure 3 – probably unnecessary.

We respectfully disagree with the referee's comment on the content of Supplementary information. While the tables and Figure S1 may not add much to the paper itself, we believe that some readers may find those interesting like we would. Given that they are in SI, we prefer to keep that as is.

**Text edits with line numbers:**

Line 15: The sentence is long and contains a lot of information, could it be rewritten as two sentences to make it clearer?

To answer a previous comment made by this referee about the abstract, we restructured the abstract to separate the results of both components of the study more clearly. This way, long sentences are now shorter.

Line 71: hence and hence and hence – too many hences.

There were indeed several 'hences'. We rephrased these sentences, lines 78-80.

Line 224: '… was more degraded than…'

We changed 'degradable' to 'degraded', line 239.

**Referee # 2**

The study assesses spatial variation in organic matter (OM) decomposition rates in peatland pools at a temperate peatland site. The authors also evaluated the influence of OM chemistry on decomposition rates of fresh litter in situ and on sediments ex situ. The authors' finding that decomposition rates vary spatially in peatland pools is highly relevant for assessments of peatland greenhouse gas (GHG) emissions and the paper will be of interest to the readership of Biogeosciences.

Thank you for the thoughtful and constructive review. Attached, we provide a point-by-point answer to the concerns you have raised.

The research approach and methods are well-designed overall, however the authors should justify the decision to oven-dry litter samples for the in situ decomposition experiment, and compare decomposition rates of litter to other studies in the discussion.

The litters were oven dried at 40°C to ensure that a uniform residual moisture content existed, particularly for *Sphagnum* which is difficult to dry uniformly and that desiccation would discourage *Sphagnum* from regrowing. We now specify this at line 127.

Regarding comparison with other studies, we added a paragraph to the discussion in which we addressed this and another comment made by the Referee. Lines 463-472.

The threshold for statistical significance should be clarified in the methods.

The significance level we used is $P < 0.05$ (except in Table 4, where we used either $P < 0.05$ or $P < 0.1$ but this is specified). However, we refrained from using the terms 'significant' in most of the text (except in three occasions where P-values were $> 0.2$) and rather chose to give the reader every $P$-value and, ultimately, the opportunity to interpret the significance of each statistic. We therefore believe it is not necessary to clarify the significance level in the methods section.

The paper presents a comprehensive dataset and the interpretation and discussion of results needs to be re-examined and strengthened in some instances. A few issues stand out:

According to the introduction OM decomposition is generally faster in pools than in surrounding soils, but in this study k values tended to be much higher in the acrotelm compared to the pools for T. latifolia and k values appear to be similar for pools and the acrotelm in the case of S. capillifolium (Table 1). This should be discussed.

Here, we are not totally sure what the referee refers to. In the introduction, we say that pools 'generally have a negative C balance because of distinct environmental conditions (e.g. higher oxygen – $O_2$ – availability and warmer temperature) compared to the surrounding soils that lead to faster OM decomposition than production'. We do not imply that decomposition is faster in the pools than in the soil, but rather that decomposition occurs faster than accumulation. To avoid any confusion, we will rephrase to: "[...] that lead to rates of OM decomposition that are faster than rates of OM production". Line 38.

We also believe the differences in decomposition rates for *Typha* and *Sphagnum* in the acrotelm vs. in the pools reinforce our justification to use both an easily degradable (*Typha*) and a more recalcitrant substrate (*Sphagnum*) to distinguish the effects of environmental and chemical

properties on litter decomposition rates. We now specify in the methodology (lines 121-126) why we chose these two litters in our experiment.

While the authors propose that decomposition rate in pools depends primarily on environmental conditions (i.e. depth), there was no difference in decomposition rates of S. capillifolium among depths (Table 1). It seems there are dual influences of environment and litter composition on decomposition rates *in situ* that the authors should consider more carefully.

Yes, there are environmental and litter quality influences at play, and a core objective of our study was to disentangle to what extent each aspect plays a role. We found that where the litter decomposition rate is 'relatively' fast (*Typha*), environment does play a role in differentiating rates. Where it is 'slow' over the short period (27 months; i.e. *Sphagnum*) patterns are not clear though they may have appeared if mass loss was larger.

We now emphasize the influence of environmental conditions (i.e. pool depth) in controlling decomposition rates in the paragraph we added at lines 463-472.

Regarding the influence of OM chemistry on decomposition of sediments ex situ statements in the abstract and discussion contradict each other. The abstract states that $CO_2$ production by sediments decreases with increasing OM humification and at line 440 it is stated that "$CO_2$ production was positively related to an increase in OM humification". The abstract states that $CH_4$ production decreases with increasing N:P but this result is not significant at $p < 0.05$ (Table 4), though there is a significant correlation between $CH_4$ and C content as well as Na content at $p < 0.05$ (see comment above about clarification of threshold for determination of statistical significance). This section of the discussion doesn't address the potential mechanisms explaining these correlations, though an explicit aim of the study is to "assess the role OM chemistry plays on decomposition rates for litter and sediments" and an implied aim is to increase knowledge of mechanisms controlling OM decomposition in peatland pools.

Thank you for pointing to what we stated at line 440 (now 484). This is clearly a typo, as shown by the results in Table 4. We have corrected this mistake in the revised ms.

We agree that there were inconsistencies between our discussion and how we present results in the abstract and how to identify potential drivers of decomposition processes in peatland pools. To address this, we changed the last sentence of the abstract to: "When combining both experiments, our results show that OM decomposition in peatland pools is highly variable and mostly related to the environmental conditions in which it occurs and as a function of general pool depth rather than to OM chemistry." Lines 24-26.

It is also true that one of our main objectives is not entirely met. We addressed this in the revised ms by adding, at section 4.2 (lines 493-500): "One of our main objectives was to assess the role OM chemistry plays on decomposition rates and to increase knowledge of the mechanisms controlling OM decomposition in peatland pools. However, because we focused our work on a small number of pools, we can only point to possible drivers of OM decomposition. For example, as suggested by the negative correlation between $CO_2$ production and humification indices or the positive correlations between C emissions and sediment P concentration (Figure 6, Table 4), it is possible that decomposition rates in peatland pools are in part driven by OM quality and P limitation. This emphasizes the lack of available information on peatland pool biogeochemical

patterns and processes and calls for further studies on the specific mechanisms responsible for sediment degradation in such unique freshwater systems."

The abstract states that differences in fresh litter and pool sediment decomposability is a function of $O_2$ concentrations, light, and temperature, which all decrease with increasing depth. Variation in these environmental parameters with depth are not presented in the present study but it seems they were included in a previous paper (Aresenault et al. 2018). The relationship between decomposition rates and $O_2$, light, and temperature should be presented in the present study to substantiate this conclusion.

This is an interesting suggestion that could greatly improve the message we are sending in the paper. Unfortunately, we only measured such parameters on 6 occasions during the 27-months period of our field work (at t=0 and then at every litter retrieving time, to be precise). Hence, we do not have a large enough sample size on which to base correlations. We must then rely on other studies that address the influences and spatial variations of $O_2$, light and temperature, one of which (Arsenault et al. 2018) was conducted at the same study site but on different pools.

To address this, we removed the mention of these specific drivers in the abstract because we did not directly measure them and added, at lines 421-426: "Unfortunately, we only measured these variables in our pools when installing and retrieving our litterbags and our sample size (n = 6) is thus too small to statistically indicate correlations between environmental conditions (i.e. $O_2$ concentration, temperature, and light) and OM decomposition. However, based on the literature as outlined in the following paragraphs, we here argue that the differences in the decomposition rates we measured were largely driven by within-pool variations in such environmental conditions that could influence the structure of microbial communities and the magnitude of decomposition processes."

It's also not clear how sediment decomposition rate can vary with depth since the sediments were collected from the bottoms of the pools. Perhaps the authors mean overall pool depth, but the ex situ decomposition rates were highest at the intermediate pool depth, not the shallowest pools. It's interesting that the total ex situ production of $CO_2$ corresponds to the sediments from the pools with the highest $CO_2$ fluxes in situ. The authors have focused on the influence of physical factors but it seems that chemistry is also important.

We indeed meant general pool depth, this is now clarified throughout the revised ms.

Our incubation study shows that decomposability (or decomposition potential) is indeed higher in the intermediate pool depth group (~1 m deep) than elsewhere, but our chemical analyses of individual sediment pool samples do not point to many correlations that could explain such results (see Table 4). We also believe that sediment chemistry is important in influencing sediment decomposition, but our experiment being the first to directly study decomposition processes in peatland pools, it is thus difficult to put forward hypotheses without falling into simple speculations.

This is why we call, at line 448 of the submitted ms (now line 499), "for further studies on the specific mechanisms responsible for sediment degradation". We believe the changes we have made in the revised ms now make clearer the hypothetical nature of the mechanisms we identify as drivers of OM decomposition in peatland pools.